# Simplifying complex machine learning by linearly separable network embedding spaces

## Abstract

Low-dimensional embeddings are a cornerstone of modelling and analysis of complex networks. However, most of the existing approaches for mining network embedding spaces rely on computationally intensive machine learning systems to facilitate downstream analysis tasks. In contrast, in the field of Natural Language Processing, it was observed that word embedding spaces capture semantic relationships **linearly**, allowing for information retrieval using simple linear operations on word embedding vectors. Similar linear semantic relationships (i.e., the compositionality of embedding vectors) have also been observed in data embeddings from pre-trained vision-language models. This poses the question of why in some cases the embedding methods lead to a linearly separable embedding space amenable to linear exploitation, while in other cases they do not. Here, we gain fundamental insight into the structure of network data that yield this linearity. We show that the more homophilic the network representation, the more linearly separable the corresponding network embedding space, yielding better downstream analysis results. We demonstrate applicability of our insight on thirteen networks from multiple domains, six multi-label biological networks and seven single-label networks from social, citation, and transportation networks domain. We believe that these fundamental insights into the structure of network data that enable their linear mining and exploitation are the foundation to build upon towards efficient and explainable mining of complex network data.

## 1 Introduction

Networks naturally model complex systems in many real-world applications. Examples include social, information, and biological domains. Current state-of-the-art approaches for analyzing these complex data are based on graph representation learning (also called network embedding) (Khoshraftar & An, 2024; Xia et al., 2026). These methods map a network's nodes in a low $d$-dimensional space, where the geometry of the space reflects the similarities between the nodes in the sense that two nodes are defined to be *similar* either when they belong to the same network neighbourhood, or have similar topological roles independent of being adjacent, e.g. being hub nodes (also called *topological similarity*). After embedding, the vectorial representation of the nodes (termed *embedding vectors*) are then fed to computationally intensive machine learning (ML) systems to perform tasks, such as node classification, clustering, and link prediction. Interestingly, in another domain, in the field of Natural Language Processing (NLP), the Skip-Gram neural network (NN) based word embeddings were shown to capture semantic relationships between words linearly, allowing for downstream analysis tasks using simple linear operations on word embedding vectors (e.g. *Word2vec* (Mikolov et al., 2013)). Understanding the emergence of such linear structure—most famously exemplified by word analogies—remains an active area of research (Arora et al., 2018; Korchinski et al., 2025). More recently, linear semantic relationships (i.e., the compositionality of embedding vectors) have also been observed in embedding spaces of pre-trained vision-language models (VLMs) (Trager et al., 2023). In the context of Language Models, this phenomenon has been formalized as the *linear representation hypothesis* (Park et al., 2024), which posits that high-level concepts (i.e., abstract semantic features learned by the model) are represented as linear directions in the representation space of a model. Empirical evidence from models such as LLaMA-2 (Touvron et al., 2023) demonstrates that these linear representations enable model

interpretability through linear probing and controllability through activation steering (Park et al., 2024). The underlying properties of the data that enable such linear compositionality remain poorly understood, raising the question of why in some cases, the embedding methods lead to a linearly separable embedding space amenable to linear exploitation, while in other cases they do not. This motivates us to explore: (i) if there is an intrinsic property in the structure of the data that yields this linearity and (ii) if the linearity holds in the biological network domain, for embeddings of the systems-level molecular networks. If so, this would simplify the analyses by making them linear, hence alleviating the need for computationally intensive ML models.

Network embedding is challenging, as it involves capturing both structural (topological) and semantic information of a graph (i.e., node labels) (Li et al., 2022) to generate low-dimensional node representations. Based on the distribution of the node labels, the networks are characterized as *homophilic* when nodes with similar labels tend to be adjacent in the network and *heterophilic* otherwise (for details, see Khoshraftar & An (2024)). Standard graph-representation learning methods, whether based on Graph Neural Networks (GNNs) (Kipf & Welling, 2017) or random walks combined with Skip-Gram architectures (Perozzi et al., 2014; Grover & Leskovec, 2016), heavily rely on the assumption of homophily. When applied to *heterophilic* networks, the performance of vanilla GNNs and neighborhood-based random walks deteriorates significantly (Alon & Yahav, 2021; Khoshraftar & An, 2024). Improving the performance of downstream analysis tasks in heterophilic data is an active area of research (Wu et al., 2025; Chen et al., 2026), yielding many increasingly complex architectures. In contrast to the design of more complex GNN architectures, researchers recently proposed to rewire the input networks in a pre-processing step by utilizing the node labels to prune heterophilic edges and add homophilic ones (Bi et al., 2024; Bose et al., 2025). These data-centric approaches nearly double the accuracy of various GNN architectures (including GCN and GAT) on heterophilic benchmarks, which suggests that restructuring data representations may be more effective than increasing model complexity.

In the context of network biology, the data can also be heterophilic, since genes with similar functions (i.e., similar annotations) can be located in distant network neighborhoods yet share similar local wiring patterns, i.e., they can be topologically similar (Milenković & Pržulj, 2008b; Malod-Dognin et al., 2019; Xenos et al., 2021). These local wiring patterns are quantified using graph substructures, which have been recognised as essential to mining complex networks—from cliques in Protein–Protein Interaction (PPI) networks to triangle counts in social networks (Leskovec et al., 2008; Bianconi et al., 2014). Such substructures are best captured by *graphlets* (Pržulj et al., 2004), small, connected, non-isomorphic, induced subgraphs of a network. Graphlets have been leveraged to generate embeddings that preserve the topological similarity of nodes (Dutta & Sahbi, 2018; Rossi et al., 2018). Graphlet-based measures have also been used to perform random walks between similarly wired nodes, independent of whether those nodes are adjacent (Xenos et al., 2021), enabling the prioritisation of novel human cancer-related genes through linear operations on gene embedding vectors. These topologically-aware random walks may therefore be more suitable for heterophilic networks. To summarise: random-walk based embeddings preserve the neighbourhood-based similarity of nodes, while graphlet-based embeddings preserve their topological similarity. Furthermore, depending on the type of similarity preserved in the embedding space, different downstream analysis tasks (e.g., predicting protein/gene function or identifying cancer-related genes) can be performed by applying simple linear operations on the node embedding vectors. This raises the question of whether leveraging both neighbourhood and topological similarities could yield embedding spaces that enable versatile downstream analyses tasks to be performed by simple linear operations on the node embedding vectors. Building on the data-centric direction of Bi et al. (2024) and Bose et al. (2025), we propose a principled, graphlet-based method for generating more homophilic representations of network data, yielding embeddings that are more linearly separable and thereby improving downstream analysis. However, rather than relying on node labels to heuristically prune heterophilic edges, we introduce a purely structural approach that preserves the graph topology. Specifically, we represent the input network through complementary higher-order adjacency matrices that capture complex topological similarities without altering the underlying data. By identifying the structural properties that enable linear exploitation of network data, we aim to pave the way for explainable and computationally efficient models.

In particular, we make the following contributions:

- **More homophilic graphlet-based network matrix representations:** We introduce novel, graphlet-based, random-walk matrix representations that account for both network neighborhood similarity and topological similarity of nodes. We apply them to represent thirteen networks from multiple domains (six multi-label biological networks and seven single-label networks from social, citation, and transportation networks domain) and we demonstrate that there always exists a graphlet-based matrix representation that is more homophilic than the standard adjacency matrix representation.

- **Linking homophily of the input matrix network representation to linear separability of the embedding space:** the abundance of these network matrix representations allows us to systematically investigate the relationship between the homophily of the network matrix representation and the linear separability of the resulting embedding space. We show that more homophilic input network matrices yield more linearly separable embedding spaces, diminishing the need for complex machine learning models in downstream tasks.

- **Improved downstream performance on heterophilic networks:** we also demonstrate that in the case of heterophilic networks, where the resulting embedding spaces are not linearly separable, our graphlet-based embeddings outperform the state-of-the-art random-walk based node embeddings, LINE and DeepWalk, by at least 8 % in the node classification F1-scores.

## 2 Preliminaries

### 2.1 Network embeddings as matrix factorization

We recall that a network is a pair, $N = (V, E)$, where $V$ is the set of nodes and where two nodes, $u$ and $v$ in $V$, can be connected by an edge $(u, v)$ in $E$. Nodes connected by an edge in $E$ are said to be adjacent or neighbors. A network is standardly represented by its adjacency matrix, $A$, in which cell $A(u, v) = 1$ if edge $(u, v)$ is in $E$, and 0 otherwise.

Recently, Qiu et al. (2018) showed that the Skip-Gram based network embeddings that rely on random walks, such as DeepWalk (Perozzi et al., 2014), LINE (Tang et al., 2015) and node2vec (Grover & Leskovec, 2016), are implicitly factorizing a random-walk-based positive pointwise mutual information (PPMI) matrix, whose cells quantify how frequently two nodes, $u$ and $v$, of the network co-occur in a random walks compared to what would be expected if the occurrences of the nodes were independent. The matrix closed formula of DeepWalk (Qiu et al., 2018) is defined as:

$$\text{DeepWalk} = \max\left(0, \log\left(vol(A)\left(\frac{1}{T}\sum_{r=1}^{T}(D^{-1}A)^r\right)D^{-1}\right) - \log b\right), \tag{1}$$

where *vol(A)* is the volume of the network, i.e., it's number of edges and is computed as $vol(A) = \sum_i \sum_j A(i, j)$, $A$ is the adjacency matrix of the network, $D$ is the diagonal matrix of degrees of the given network, in which each diagonal element $D(i, i)$ is the sum of all the entries in row $i$ of $A$, *T=10* is the length of the random walks and $b$ is the negative sampling in Skip-Gram. For computational reasons, in the study of Qiu et al. (2018), they set $b = 1$ to omit the constant term $-\log b$. They demonstrate that the explicit factorization of the DeepWalk closed matrix formula with Singular Value Decomposition (SVD) leads to equivalent or better performance in network mining tasks than the implicit Skip-Gram based ones. Subsequently, it was demonstrated that the factorization of Deepwalk closed matrix formula with Non-Negative Matrix Tri-Factorization (NMTF), a popular matrix factorization approach used notably for integrating (fusing) and mining networks, could achieve equivalent or better results in node classification tasks in biological networks by using simple linear operations on the node embedding vectors (Xenos et al., 2021). The key difference between NMTF and SVD is that the resulting matrix factors of NMTF are non-negative, which allow for their intuitive and easy interpretation (Devarajan, 2008; Hao et al., 2021). This feature makes NMTF particularly effective for molecular and clinical data, where explainability and interpretability are essential (Combi et al., 2022).

LINE is a special case of the DeepWalk using random walks of length, $T = 1$. This type of random walk considers that two nodes can co-occur only if they are directly connected by an edge. The closed formula of LINE (Qiu et al., 2018) is defined as:

$$\text{LINE} = \max\left(0, \log\left(vol(A)D^{-1}AD^{-1}\right) - \log b\right). \tag{2}$$

In this study, to generate more homophilic matrix representations that capture simultaneously topological and neighborhood based similarity, we propose to extend LINE and DeepWalk closed formulas by using graphlets as context for the random walks.

## 2.2 Graphlets and Graphlet Adjacency

*Graphlets* are small, connected, non-isomorphic, induced sub-graphs of a large network that appear at any frequency (Pržulj et al., 2004). Different topological positions within graphlets are characterized by different symmetry groups of nodes, called *automorphism orbits* (Pržulj, 2007). Orbits describe the different topological roles of a node in a particular graphlet. The nine 2- to 4-node graphlets are denoted by $G_0, \ldots, G_8$ and the corresponding fifteen orbits by $o_0, \ldots, o_{14}$ (see Figure 3). A widely used node-level descriptor that summarize the wiring of a node in a network is the *Graphlet Degree Vector (GDV)*, which captures the frequency of a node appearing in each automorphism orbit. By comparing the GDVs of two nodes, a measure called *GDV similarity* is defined that quantifies how topologically similar two nodes are (detailed in Appendix A.2). The pairwise GDV similarities of all nodes are represented in the *GDV similarity matrix*, which captures the pairwise topological similarities of the nodes. Recently, the GDV similarity matrix was used as the input for the DeepWalk closed formula, i.e. the GDV similarities were used as transition probabilities for the random walks, to generate embeddings that are based solely on the topological similarities between the nodes (GDV PPMI) (Xenos et al., 2021). However, these embeddings are limited by the fact that do not account for neighborhood information, which is known to be important for complex networks.

The original definition of the graphlet-based measures quantify the local wiring patterns around network nodes. However, these measures do not consider whether two nodes participate in the same network neighborhood (i.e., in the same subgraph). To bridge the gap between local neighborhood and topological similarity, Windels et al. (2019) introduced the concept of the *graphlet adjacency*, in which a pair of nodes are "graphlet-adjacent" if they simultaneously touch (participate in) a given graphlet. Hence, each graphlet spans a corresponding Graphlet Adjacency. For higher-order graphlets, two nodes can be adjacent more than once. For instance, between two nodes may exist more than one 3-node path that connects them. Given this extended definition of adjacency, for each graphlet $G_k$, the corresponding graphlet adjacency matrix $A_{G_k}$ is defined as $A_{G_k}(u, v) = a_{uv}^k$ if $u \neq v$, or 0 otherwise, where $a_{uv}^k$ is equal to the number of times nodes $u$ and $v$ are graphlet-adjacent with respect to a given graphlet $G_k$.

# 3 Methodology

## 3.1 Novel graphlet-based network matrix representations

To extend the PPMI formulas used in network embedding to consider local graph substructures, we propose to use as the corpus all the instances of a given graphlet in the network and then to compute how frequently two nodes co-appear in the instances of a given graphlet. Recall that a Graphlet Adjacency matrix encodes in how many instances of a given graphlet two nodes co-occur. Thus, to transform the raw co-occurrences to probabilities of co-occurrences, we apply to each Graphlet Adjacency matrix the PPMI formulas of DeepWalk and LINE (see equation 1 and 2 defined above). Note that to generate the graphlet adjacency matrices, we use the implementation in Windels et al. (2019).

In the **LINE closed formula**, the key parameter is the adjacency matrix, $A$, which captures wether two nodes are directly connected by an edge (i.e., co-occure on random-walks of length one). To incorporate the graphlets in the LINE closed formula we propose to consider that two nodes co-occure if they are graphlet-adjacent rather than being solely connected by an edge. Formally, for a each graphlet $G_k$, our new

graphlet-based extension of LINE, which we denote by "$\text{LINE}_{G_k}$," is defined as:

$$\text{LINE}_{G_k} = \max\left(0, \log\left(vol(A_{G_k})D_{G_k}^{-1}A_{G_k}D_{G_k}^{-1}\right)\right), \tag{3}$$

where $vol(A_{G_k})$ is the volume of the graphlet adjacency matrix $A_{G_k}$, i.e., the sum of all its entries, and $D_{G_k}$ is the diagonal degree matrix of $A_{G_k}$, where each diagonal element $D_{G_k}(i,i)$ is the sum of all the entries in row $i$ of $A_{G_k}$.

In the **DeepWalk closed-formula**, the key parameter is the adjacency matrix of the network, $A$, in which the random walks of length $T$ are computed. Hence, to incorporate the graphlets in the DeepWalk closed formula, we propose to compute random walks in all the different nine graphlet adjacency matrices. In a graphlet adjacency matrix, $A_{G_k}$, the entries represent the number of times two nodes simultaneously participate in the given graphlet. This number is not bounded and can be very big especially for higher-order graphlets. To address this, we introduce the binarized graphlet adjacency matrix, $\tilde{A_{G_k}}$, by setting all the non-zero values to one. Then, we use as input for the DeepWalk closed-formula any of the nine binarized graphlet adjacencies. Formally, for a each graphlet $G_k$, our new graphlet-based extension of DeepWalk, which we denote by "$\text{DeepWalk}_{G_k}$," is defined as:

$$\text{DeepWalk}_{G_k} = \max\left(0, \log\left(vol(\tilde{A_{G_k}})\left(\frac{1}{T}\sum_{r=1}^{T}(\tilde{D_{G_k}}^{-1}\tilde{A_{G_k}})^r\right)\tilde{D_{G_k}}^{-1}\right)\right), \tag{4}$$

where $vol(\tilde{A_{G_k}})$ is the volume of the binarized graphlet adjacency $\tilde{A_{G_k}}$, $\tilde{D_{G_k}}$ is the diagonal degree matrix of $\tilde{A_{G_k}}$ and $T{=}10$ is the length of the random walks.

In both proposed extensions, we will have nine different graphlet-based network representations that account for the different connectivity patterns captured by each of the nine 2-node to 4-node graphlets. Note that for graphlet $G_0$, the graphlet-adjacency matrix $A_{G_0}$ corresponds to the original adjacency matrix, $A$. Hence, $\text{LINE}_{G_0}$ and $\text{DeepWalk}_{G_0}$ correspond to the original LINE and DeepWalk, respectively.

### 3.2 Homophily measures

Standard homophily measures are defined for a network, $N = (V, E)$. To apply these measures on the different matrix representations of a network, we observe that because each network matrix representation captures a different notion of what it means for two nodes to be adjacent, it can be interpreted as defining a different set of edges for the considered network. Formally, for a network, $N = (V, E)$, the matrix representation, $M$, defines a new set of edges, $E_M$, in which two nodes, $u \in V$ and $v \in V$, are connected by an edge $(u, v)$ in $E_M$ if $M(u, v) \neq 0$. The adjacency and the binarized graphlet adjacency matrices are unweighted, as they only consider the nodes to be adjacent or not. On the other hand, LINE, DeepWalk and our graphlet-based extensions are weighted, with higher values in the matrix representations indicating stronger connections between the nodes. For such weighted matrix representations, each edge $(u, v)$ in $E_M$ is associated with a weight, $w_{uv} = M(u, v)$.

To measure if the unweighted matrix representations of a network are homophilic (adjacent nodes share the same label), we use the two standard metrics: the edge homophily index and node homophily index (Zhu et al., 2020). Formally, for a given matix representation, $M$, the edge homophily index, $H_{edge}(M)$, quantifies the proportion of edges in $E_M$ connecting two nodes from the same class and is defined as:

$$H_{edge}(M) = \frac{|(u, v) \in E_M : label(u) = label(v)|}{|E_M|}.$$

The node homophily index, $H_{node}(M)$, quantifies the average proportion of the adjacent nodes that share the same label (class) (Zheng et al., 2022). Formally, it is defined as:

$$H_{node}(M) = \frac{1}{|V|}\sum_{u \in V}\left(\frac{|v \in N(u) : label(u) = label(v)|}{|N_M(u)|}\right),$$

where $N_M(u)$ is the set of the neighbors of node $u$ according to the edges in $E_M$.

We also extended these indices for the weighted matrix representations of a network to assess if nodes with stronger connections (i.e., higher edge weights) share the same label. If two nodes share the same label, we use the corresponding edge weight instead of 1. The weighted edge homophily index is defined as:

$$H_{\text{edge}}^{\text{weighted}}(M) = \frac{\sum_{(u,v) \in E_M} w_{u,v} : \text{label}(u) = \text{label}(v)}{\sum_{(u,v) \in E_M} w_{u,v}}.$$

The weighted node homophily index is defined as:

$$H_{\text{node}}^{\text{weighted}}(M) = \frac{1}{|V|} \sum_{u \in V} \left( \frac{\sum_{v \in N_M(u)} w_{u,v} : \text{label}(u) = \text{label}(v)}{\sum_{v \in N_M(u)} w_{u,v}} \right),$$

where $N_M(u)$ is the set of the neighbors of node $u$ according to the edges in $E_M$.

We also compute the Geometric Separability Index (GSI) (Thornton, 1998), a less stringent version of the node homophily index that can also be applied to weighted matrix representations of networks. GSI is defined as the proportion of network nodes whose labels are the same as those of their first-nearest neighbor. Formally, given a weighted network matrix representation $M$, the GSI is defined as:

$$\text{GSI}(M) = \frac{1}{|V|} \sum_{u \in V} \mathbb{I}(label(u) = label(\text{NN}_u)),$$

where $\mathbb{I}(\cdot)$ is the indicator function and $\text{NN}_u$ is the nearest neighbor of node $u$. If the network is weighted, the nearest neighbors are identified using the edge-weights in $M$. Otherwise, GSI uses as edge-weights the Euclidean distances between the rows of $M$. We also extend GSI, node and edge homophily indices (weighted and unweighted) for multi-label (biological) networks in which a node may have more than one annotation. To do so, for each node, we compute the fraction of shared annotations (labels) with its nearest neighbor.

### 3.3 Linear separability of the embedding space

We generate the network embedding spaces by factorizing all the different matrix representations (adjacency, LINE, DeepWalk and all their graphlet-based extensions) of the single-label and multi-label networks using the ONMTF framework (detailed in Appendix A.3). An embedding space is considered linearly separable when the embedding vectors of nodes from different classes can be separated by hyperplanes. If the classes in the embedding space are linearly separable, then a linear classifier can classify the nodes into their respective classes as accurately as a non-linear classifier. Hence, to assess the linearity of a space, we perform node classification on single-label and multi-label networks using Support Vector Machines (SVM) with both linear (Euclidean kernel) and non-linear (Radial Basis Function (RBF) kernel) approaches. In addition, we compare their performance with that of the Random Forest (RF) classifier, a state-of-the-art non-linear method (Breiman, 2001). To evaluate the accuracy of the classifiers, we use 10-fold cross-validation and compute the corresponding weighted F1-score for each classifier.

For our analysis, we term a space *fully linearly separable* if the node classification F1-scores of the linear SVM (euclidean kernel) is greater than or equal to 0.8. If the node classification F1-scores of the linear SVM is lower than 0.8 but is on par or better than the F1-score of the non-linear SVM (RBF kernel), we term the space as *sufficiently linearly separable*, otherwise, we term the space as *non-linear*.

## 4 Experimental Results

### 4.1 Data

**Biological multi-labeled networks:** we use the most well-studied molecular networks, modelling protein-protein interactions and gene co-expressions of the following three species: *Homo sapiens* (human), *Saccharomyces cerevisiae* (budding yeast) and *Schizosaccharomyces pombe* (fission yeast). We collected the experimentally validated protein-protein interactions (PPIs) from BioGRID v.3.5.182 (Oughtred et al.,

2021) and the gene co-expressions (COEX) from COXPRESdb v.8 (Obayashi et al., 2022) (for details on how we constructed the six biological network and an overview of their statistics see Appendix A.4). As node class-labels, we use the level 1 Gene Ontology Biological Process (GO BP) terms (The Gene Ontology Consortium, 2025), which represent higher-level biological functions.

**Single-labeled networks:** We analyze single-labeled networks, beyond biology, that are used as benchmarking datasets in ML studies. We collected from PyTorch Geometric (Fey & Lenssen, 2019) the USA air-traffic network (Ribeiro et al., 2017), the Coauthor CS network (Shchur et al., 2018), the CORA and the CiteSeer citation networks (Sen et al., 2008), the two standard heterophilic Wikipedia page–page networks (Chameleon and Squirrel) (Rozemberczki et al., 2021) and the Wiki-CS hyperlinks network (Mernyei & Cangea, 2020). For all the datasets, we treat the networks as undirected and only consider their largest connected component. The statistics of the aforementioned networks are presented in Appendix Table 5.

## 4.2 Graphlet random walk-based matrices yield more homophilic network representations

To demonstrate that our new graphlet-based extensions of LINE and DeepWalk yield more homophilic representations of the networks than the original baseline methods (that correspond to our $G_0$ extension, as detailed in Section 3.1), we measure the level of homophily of the different matrix representations of six molecular multi-label networks and seven single-label networks (detailed above). Traditionally, in unweighted networks represented by their standard adjacency matrices, homophily is measured by the edge homophily index and the node homophily index (detailed in Section 3.2) (Zheng et al., 2022). In weighted networks, it is measured by the GSI (Thornton, 1998), a simplified measure that compares the label of a node only with the label of its closest neighbor. In this study, we extend these measures to quantify the homophily/heterophily in any matrix representation of the network (weighted or unweighted). In the case of the molecular networks, we annotate the genes with level 1 Gene Ontology Biological Process (GO BP) terms (The Gene Ontology Consortium, 2025), which represent higher-level biological functions. Since the functions (annotations) of all genes are not known, we report the results over the set of genes with known annotations. For the number of annotated nodes in each of these networks, see Appendix Table 6.

We observe that for both the multi-label and the single-label networks, there is at least one higher-order graphlet-based network representation that is more homophilic (i.e., having greater GSI, node homophily and edge homophily indices) than the baseline adjacency matrix, $\text{DeepWalk}_{G_0}$ and $\text{LINE}_{G_0}$ (see Figure 1). In addition, over all networks and over all graphlets, our new $\text{DeepWalk}_{G_k}$ and $\text{LINE}_{G_k}$ graphlet-based representations that capture both topological and neighborhood-based similarity yield more homophilic representations than GDV similarity matrix and GDV PPMI matrix that are based solely on topological similarity (see Figure 1). Among the different network representations, $\text{LINE}_{G_2}$, $\text{LINE}_{G_8}$, $A_{G_2}$ and $A_{G_8}$ lead to the most homophilic representations in terms of the node and edge homophily index (see Figure 1 and Supplementary Figure 4). In addition, our new $\text{DeepWalk}_{G_k}$ yield the most homophilic representations in terms of the GSI (see Panel C and D of Figure 1): $\text{DeepWalk}_{G_1}$ in case of single-label networks, and $\text{DeepWalk}_{G_2}$ in case of multi-label networks.

In summary, for both the multi-label and the single-label networks, our new graphlet-based network representations are more homophilic than the original representations that capture either only the direct neighbourhood similarity, or only the topological similarity. In the following sections, we assess if embedding the networks by factorizing these more homophilic matrix representations leads to more linearly separable network embedding spaces.

## 4.3 Graphlet random walk-based network representations lead to linearly separable embedding spaces

We examine if our graphlet-based extensions of the state-of-the-art random-walk based node embedding methods yield more linearly separable network embedding spaces than the original baseline methods (that also correspond to our $G_0$ extension). That is, we examine if they can be applied to result in embeddings of nodes from different classes that can be separated by hyperplanes. To do that, first we generate the network embedding spaces by factorizing all the different matrix representations ($\text{DeepWalk}_{G_k}$, $\text{LINE}_{G_k}$ and $A_{G_k}$) of the seven single-label networks by utilizing the ONMTF framework (see Appendix A.3). Recall that

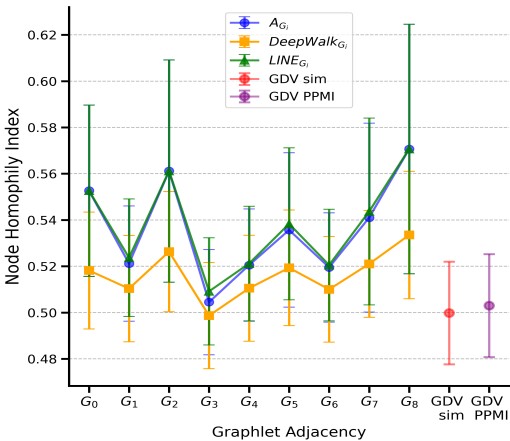 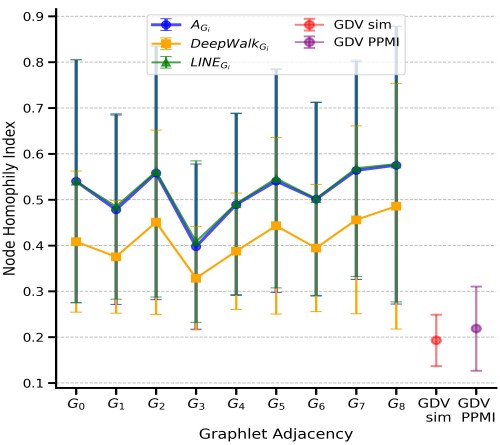

Figure 1: Graphlet-based network matrix representations lead to more homophilic representations. In the left Panel, for each graphlet (x-axis) and method (color-coded), the line plot shows the average, over the six biological multi-label networks, node homophily index and the standard deviation. In the right Panel shows the same, but on average over the seven single-label networks.

| Networks | DeepWalk$_{G_k}$ | | | LINE$_{G_k}$ | | | Graphlet Adjacency, A$_{G_k}$ | | |
|---|---|---|---|---|---|---|---|---|---|
| | L-SVM | SVM RBF | RF | L-SVM | SVM RBF | RF | L-SVM | SVM RBF | RF |
| Cora | 0.825 | 0.821 | **0.836** | 0.767 | 0.756 | 0.806 | 0.721 | 0.756 | 0.783 |
| Wikipedia CS | **0.821** | 0.820 | 0.818 | 0.776 | 0.771 | 0.778 | 0.703 | 0.693 | 0.763 |
| CS Co-author | 0.901 | 0.894 | **0.906** | 0.849 | 0.833 | 0.864 | 0.809 | 0.783 | 0.853 |
| USA air-traffic | 0.655 | 0.637 | 0.668 | 0.670 | 0.671 | **0.702** | 0.647 | 0.642 | 0.696 |
| Chameleon | 0.679 | 0.641 | **0.768** | 0.699 | 0.670 | 0.754 | 0.691 | 0.648 | 0.765 |
| CiteSeer | 0.702 | 0.712 | **0.736** | 0.637 | 0.662 | 0.725 | 0.544 | 0.561 | 0.712 |
| Squirrel | 0.498 | 0.391 | 0.748 | 0.598 | 0.472 | 0.758 | 0.573 | 0.439 | **0.765** |
| Pombe COEX | 0.527 | 0.525 | 0.511 | 0.527 | **0.53** | 0.503 | 0.518 | 0.519 | 0.498 |
| Cerevisiae COEX | **0.611** | 0.595 | 0.582 | 0.603 | 0.591 | 0.578 | 0.597 | 0.575 | 0.576 |
| Homo sapiens COEX | 0.507 | **0.52** | 0.496 | 0.515 | 0.518 | 0.494 | 0.508 | 0.507 | 0.494 |
| Pombe PPI | **0.671** | 0.656 | 0.624 | 0.598 | 0.591 | 0.6 | 0.554 | 0.531 | 0.587 |
| Cerevisiae PPI | **0.648** | 0.633 | 0.627 | 0.633 | 0.624 | 0.615 | 0.607 | 0.596 | 0.599 |
| Homo sapiens PPI | **0.519** | 0.513 | 0.489 | 0.514 | 0.509 | 0.480 | 0.50 | 0.494 | 0.475 |
| Average | 0.659 | 0.643 | 0.678 | 0.645 | 0.631 | 0.666 | 0.613 | 0.596 | 0.659 |

Table 1: **Node classification F1-scores in the single-label and multi-label networks.** For each network (row), the table shows the maximum weighted node classification F1-score of the corresponding classifier (linear SVM (L-SVM), SVM RBF and RF) in the embedding spaces obtained from different matrix representations (columns). The first seven networks correspond to the single-labeled networks and the last six networks to the multi-labeled molecular networks. Note that bold cells indicate the highest value per row.

this involves nine graphlet-based matrix representations (because there are nine up to 4-node graphlets) per method (for the 3 methods, DeepWalk$_{G_k}$, LINE$_{G_k}$ and $A_{G_k}$) for each of the seven single-label networks, hence yielding the total of $9 \times 3 \times 7 = 189$ embedding spaces. Similarly, we generate the embeddings of the six molecular interaction networks (i.e., nine graphlet-based representations per method per network, yielding the total of $9 \times 3 \times 6 = 162$ embedding spaces). Recall that if the classes in an embedding space are linearly separable, then a linear classifier should classify the nodes into their respective classes as accurately as a non-linear classifier. Hence, we assess the linear separability of an embedding space produced by our representations by comparing the resulting node classification weighted F1-scores obtained by SVM with linear kernel (Euclidean) and non-linear kernel (RBF). In addition, we assess it by comparing with that of the RF classifier, a state-of-the-art non-linear method (Breiman, 2001) (see Section 3.3). We term a space as fully linearly separable if the node classification F1-scores of the linear SVM (euclidean kernel) is greater than or equal to 0.8. In addition, if the node classification F1-scores of the linear SVM is lower than 0.8 but is on par or better than the F1-score of the non-linear SVM (RBF kernel), we term the space as

*sufficiently linearly separable.* Otherwise, we term the space as *non-linear.* As shown in Table 1, in three of the networks (Cora, Wikipedia CS and CS Co-author), the corresponding embedding spaces are fully linearly separable, since the F1-scores exceed 0.8 (average weighted F1-score of 0.85, see left panel in Figure 5 in Appendix A.7) and there are no statistically significant differences between the classification results of the linear and the non-linear approaches (Mann Whitney U-test between their distributions of F1 scores having $p - values >= 0.05$). In addition, in six of the networks, the corresponding embedding spaces are sufficiently linearly separable, since the node classification F1-scores of the linear SVM outperform those of the non-linear SVM (RBF kernel). Only in four single-label networks (USA air-traffic, Chameleon, Squirrel, and CiteSeer) the embedding spaces are not linearly separabable, since the RF classifier outperforms the linear and the non-linear SVM by at least 14% in the node classification F1-scores (see right panel of Figure 5 in A.7).

Regardless of the classifier, the best results over all networks are obtained with our DeepWalk$_{G_k}$(i.e., our graphlet-based extensions of DeepWalk) with an average F1-score of 0.66, followed by LINE$_{G_k}$ (0.647 average F1-score) and then by the raw A$_{G_k}$ (0.623 average F1-score). Hence, our new DeepWalk$_{G_k}$ and LINE$_{G_k}$ representations that employ random-walks to diffuse the information on the input adjacency matrix lead to more separable embedding spaces than the $G_{Adj}$ representations that embed directly the adjacency matrices. For the three networks with the embedding spaces that are fully linearly separable, the highest F1-scores are obtained with the standard adjacency matrix representation, $G_0$ (see Supplementary Table 7). This means that the node classes are already linearly separable in the embeddings generated by the standard adjacency matrix and hence, the higher-order graphlets are not needed to disentangle them. In contrast, in three out of the six networks with sufficiently linearly separable embedding spaces , i.e., for *Pombe* COEX, *Cerevisiae* PPI and *Homo sapiens* PPI network, the highest F1-scores are achieved with our higher-order graphlet-based embeddings: LINE$_{G_3}$, DeepWalk$_{G_8}$ and DeepWalk$_{G_2}$, respectively (see Supplementary Table 7). In the four networks yielding embedding spaces that are non-linearly separable, which include the two standard heterophilic networks, our graphlet-based extensions of DeepWalk (DeepWalk$_{G_2}$, DeepWalk$_{G_4}$ and DeepWalk$_{G_6}$) yield the maximum weighted F1-score of 0.72, outperforming the original DeepWalk$_{G_0}$ method by almost 8%. Hence, in the networks with the resulting embedding spaces that are sufficiently linearly separable or non-linear (the RF classifier outperforms the linear classifiers), our higher-order DeepWalk$_{G_k}$ and LINE$_{G_k}$ lead to embeddings that better separate the different node classes.

In conclusion, we demonstrate in single-label and multi-label networks that our proposed graphlet-based embeddings lead to better class separability in the embedding space compared to the methods that rely only on topological similarity, or only on the direct neighborhood similarity. However, despite these improvements, some embedding spaces remain non-linear. Hence, in the following section, we investigate whether there is an intrinsic property in the input network matrix representation that is related to the linear separability of the resulting embedding space.

### 4.4 More homophilic network representations lead to linearly separable embedding spaces

Table 2: **Correlation between homophily levels of the different graphlet-based network representations and the linear separability in the resulting embedding spaces.** The table shows the Pearson's correlation coefficients between the homophily indexes and the node classification F1-scores obtained by using the linear SVM (L-SVM) in the embedding space of simulated networks from the random partition model (Column 2), of the seven single-labeled networks (Column 3) and of the six molecular multi-labeled networks (Column 4). The values in parenthesis are the p-values and the bold cells indicate statistical significance of the Pearson's correlation.

| Pearson's Correlation | Synthetic data | Single-label networks | Multi-label networks |
|---|---|---|---|
| Node homophily — F1-score L-SVM | **0.19** $(1.16 \times 10^{-11})$ | **0.36** $(1.86 \times 10^{-06})$ | **0.53** $(7.33 \times 10^{-13})$ |
| Edge homophily — F1-score L-SVM | **0.19** $(6.43 \times 10^{-10})$ | **0.35** $(4.16 \times 10^{-06})$ | **0.44** $(3.84 \times 10^{-09})$ |
| GSI — F1-score L-SVM | **0.68** $(5.21 \times 10^{-151})$ | **0.44** $(3.17 \times 10^{-10})$ | **0.46** $(1.8 \times 10^{-09})$ |

We hypothesize that the more homophilic the input network matrix representation, the more linearly separable are the node classes in the resulting embedding space. To demonstrate this under ideal conditions, without noisy data, we generate synthetic data from the random partition graph model (Fortunato, 2010). This model allows the generation of random graphs that exhibit clustering and modular organization akin to real-world networks. We simulate 420 networks, each containing 1,000 nodes distributed across five communities (detailed in Appendix A.5). We represent each network with the DeepWalk$_{G_0}$ closed matrix formula, the LINE$_{G_0}$ closed matrix formula and the standard Adjacency matrix. For each network's matrix representation, we compute its node homophily index, edge homophily index and GSI. Subsequently, we generate the network embedding spaces by factorizing their aforementioned matrix representation by using the ONMTF framework (detailed in Appendix A.3) and for each embedding space, we compute the node classification F1-score with the linear SVM (Euclidean kernel). We observe a positive correlation between the homophily level of the input matrix representation and the node classification F1-scores: 0.68 Pearson Correlation Coefficient (PCC) for GSI and 0.19 PCC for node and edge homophily indices, all statistically significant (see Table 2). To assess if this observation holds for our real networks, we computed the correlations between the homophily indices of the network matrix representations of our 13 real networks and the node classification F1-scores in the corresponding embedding spaces. In the single-label networks, we observe 0.44 PCC for GSI and approximately 0.36 PCC for node and edge homophily indices, all statistically significant (see Table 2). In the multi-label molecular networks, we observe 0.46 PCC for GSI, 0.53 PCC for the node homophily index and 0.44 PCC for the edge homophily index, all statistically significant (see Table 2).

Hence, we verify our hypothesis in simulated and real world networks that the more homophilic the input network matrix representation, the more linearly separable are the node classes in the resulting embedding space. In the following section, we show that our new graphlet-based matrix representations of networks, which are more homophilic than their traditional matrix representations, also yield network embeddings resulting in better results in downstream analysis tasks.

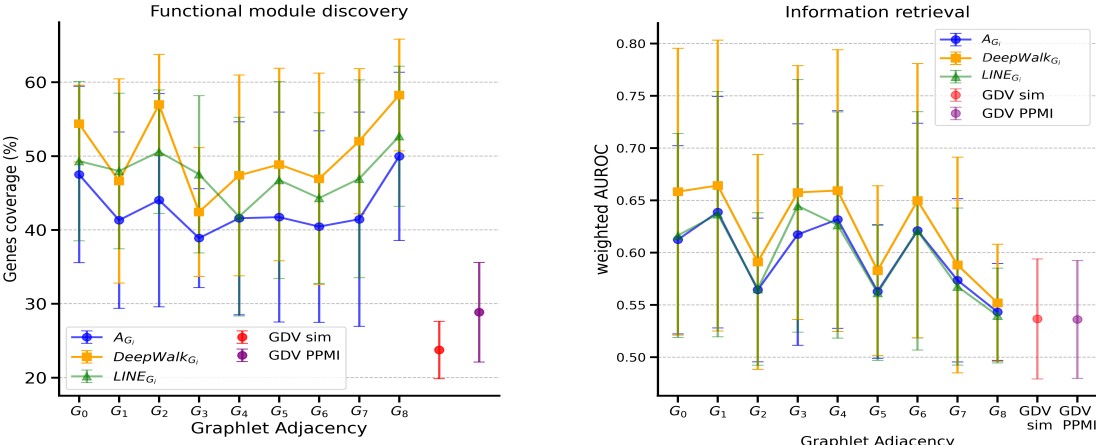

Figure 2: **Graphlet-based embeddings lead to better results in downstream analysis tasks.** Left Panel presents the results of the functional module discovery in gene embedding spaces of the biological multi-labeled networks. Right Panel presents the results of the label prediction based on the cosine similarity in the embedding space of the single-labeled networks.

## 4.5 Graphlet-based embeddings lead to better results in downstream analysis tasks

To demonstrate that higher linear separability of classes in the embedding space leads to better downstream analysis results, we perform two different experiments: one for single-label and one for multi-label real-world networks. In single-label networks, a common information retrieval task is to predict the label of a node based on the label of its most semantically similar node, defined by the largest cosine similarity of their embedding vectors. Therefore, we evaluate whether our new graphlet-based network embedding spaces improve information retrieval by assessing if nodes whose embedding vectors have high cosine similarity have

the same label. Formally, for each network, we compute its weighted area under the ROC curve (AUROC) (Bradley, 1997) between the ground truth (label) and the prediction score (cosine distance) over all the pairs of nodes in the network. For all the matrix representation families (DeepWalk$_{G_k}$ and LINE$_{G_k}$ and A$_{G_k}$), our extensions based on higher-order graphlets $G_1$, $G_3$, $G_4$ and $G_6$ lead to embedding spaces that allows better information retrieval based on the cosine similarities of the nodes than the original matrix representations that are based on the standard adjacency matrix, $G_0$ (see left Panel in Figure 2). In particular, our graphlet-based extensions of DeepWalk yield the most suitable embedding spaces for this downstream analysis task, with DeepWalk$_{G_1}$ achieving the highest weighted-AUROC (0.669 on average over the seven networks), followed by DeepWalk$_{G_3}$ (with weighted-AUROC of 0.663) and then by DeepWalk$_{G_4}$ (with weighted-AUROC of 0.659). Also, we observe that our newly introduced family of methods (DeepWalk$_{G_k}$, LINE$_{G_k}$) outperform the ones that rely only on the topological similarity, GDV similarity matrix and the GDV PPMI matrix, which have, on average, 0.33 weighted AUROC score.

In addition, we assess the performance of our graphlet-based embeddings in downstream analysis tasks in molecular, multi-label networks, in which nodes (i.e., genes) may lack annotations (labels), or have overlapping annotations (labels). A standard task is uncovering functional network modules, i.e., sets of genes that together perform higher-level biological functions. After embedding a biological network, this is typically done by clustering the genes based on their proximity (Euclidean distance) in the embedding space. Then, we assess if genes that are clustered together significantly share biological annotations by using over-representation (also called "enrichment") analysis, that accounts for the incompleteness, noisiness and overlapping of biological annotations. To perform this analysis, we annotate genes in the molecular networks with Reactome Pathway (RP) terms, GO BP terms and KEGG pathways and we report the percentage of enriched functions (*functional coverage*), which quantifies the number of functions over-represented (enriched) in some parts of the embedding space, and the percentage of enriched genes (*gene coverage*), which quantifies the number of genes coding for proteins with these enriched functions (for details, see "Functional module discovery in biological networks" in Appendix A.5). Because we used GO BP terms to define the embedding space separability (see Section 4.3), we use RP terms for the functional module discovery task to avoid circularity when correlating separability with the downstream analysis tasks. In Appendix Figures 7 and 8, we also present the results of using GO BP terms and KEGG pathways, which are consistent with those obtained using RP terms.

We find that over all molecular networks, there is always a graphlet-based extension of LINE and DeepWalk that outperforms the original method (corresponding to the adjacency matrix of the network, $G_0$) in terms of functional and gene coverages for RP terms (see right Panel in Figure 2 and Supplementary Figure 6). The best results are obtained by DeepWalk$_{G_k}$, followed by LINE$_{G_k}$ and then by A$_{G_k}$ (the Graphlet Adjacencies). This further verifies that the embeddings based on the DeepWalk$_{G_k}$ and LINE$_{G_k}$ that employ random-walks to diffuse the information on the input adjacency matrix capture more biological information than the A$_{G_k}$ family that embed directly the adjacency matrices (Xenos et al., 2021). The highest gene and functional coverage occurs with our DeepWalk$_{G_2}$ and DeepWalk$_{G_8}$, the two dense graphlet-based extensions (i.e. the 3-node and 4-node cliques) of the DeepWalk closed formula, which we have already shown to yield the most homophilic network matrix representations of molecular networks. In particular, the average gene coverage for DeepWalk$_{G_0}$ is 54.39%, whereas for DeepWalk$_{G_2}$ and DeepWalk$_{G_8}$ the average gene coverages are 56.97% and 58.95%, respectively. The better performance in functional modules discovery of these interconnected graphlets align with the biological hypothesis that functions are performed by densely interconnected genes. For instance, proteins with similar molecular functions often form protein complexes, which are represented as dense subgraphs within the PPI network (Chen & Yuan, 2006). Finally, over all molecular networks and all graphlets, our new network embedding approaches that leverage both topological and direct neighborhood-based similarity (DeepWalk$_{G_k}$ and LINE$_{G_k}$) yield embedding spaces that uncover more biologically coherent functional network modules than the embedding spaces that are based only on topological similarity (GDV similarity matrix and GDV PPMI matrix); see right Panel in Figure 2 and Appendix Figure 6. The other annotations (GO BP and KEGG pathways) show consistent results, as shown in Supplementary Figures 7 and 8 (in Appendix A.7).

In conclusion, we demonstrate that our new graphlet-based network embeddings, which are based on factorizing more homophilic network matrix representations, yield better results in downstream analysis tasks. In

the next section, we show the connection between these improved results and our more homophilic graphlet-based matrix representations of the networks.

### 4.6 More homophilic network representations lead to better results in downstream tasks

Table 3: **Correlation between homophily levels of the different graphlet-based network representations and the results of the downstream analysis tasks.** The table shows the Pearson's correlation coefficients between the homophily indexes and the results of the downstream analysis tasks in the embedding spaces of the seven single-labeled networks (Column 2) and of the six multi-labeled molecular networks (Column 3). In the single-labeled networks, the downstream analysis task is the information retrieval by using the cosine similarity; in the multi-labeled molecular networks, it is the functional module discovery. The values in parenthesis are the p-values and the bold cells indicate statistical significance of the Pearson's correlation (p-value $< 0.5$).

| Pearson's Correlation | Single-label networks | Multi-label networks |
|---|---|---|
| Node homophily - downstream analysis results | **0.61** $(1.01 \times 10^{-18})$ | **0.49** $(3.36 \times 10^{-11})$ |
| Edge homophily - downstream analysis results | **0.57** $(1.07 \times 10^{-15})$ | **0.36** $(2.71 \times 10^{-06})$ |
| GSI - downstream analysis results | **0.17** $(0.02)$ | **0.36** $(2.56 \times 10^{-06})$ |

To demonstrate that the more homophilic the input network matrix representation, the better the results of the downstream analysis tasks, we compute the Pearson's correlation coefficients (PCC) between the homophily measures, the information retrieval results (in single label networks) and the functional module discovery (in multi-label molecular networks) results. For the seven single-label networks, as shown in Table 3, all the homophily measures are statistically significantly correlated with the information retrieval task, having the following PCCs: 0.17 PCC for GSI, 0.61 PCC for node homophily and 0.57 PCC for edge homophily. For the six molecular, multi-labeled networks, all the homophily measures are also statistically significantly correlated with the functional module discovery, having the following PCCs: 0.36 PCC for GSI, 0.49 PCC for node homophily and 0.36 PCC for edge homophily.

In conclusion, we demonstrate that: (i) our graphlet-based matrix representations of the networks are more homophilic than the traditional ones; (ii) the more homophilic the matrix representation of the network, the more linearly organized the resulting embedding space, and (iii) the better the downstream analysis results in the embedding space.

## 5 Conclusions

We introduce novel network embedding methods built on graphlet-based random walk matrix representations of networks; importantly, they capture both higher-order topological and direct neighbourhood information. Our novel matrix representations, alongside the traditional ones that capture either only the higher-order topological or only the direct neighborhood-based node similarity (but not both), enable us to explore the relationship between the intrinsic properties of the given network matrix representation, the topology of the resulting embedding space and the downstream analysis results in the embedding space. We demonstrate, on synthetic and 13 real-world networks from several application domains (including six multi-labeled and seven single-labeled networks), that the more homophilic the network representation, the more linearly separable the corresponding network embedding space, yielding better downstream analysis results.

In this study, we focus on undirected and unweighted networks. However, our methodology can be easily extended and applied on any network type for which graphlets have been defined, such as for directed networks (Sarajlić et al., 2016), for weighted / probabilistic networks (Doria-Belenguer et al., 2020), hypernetworks (Gaudelet et al., 2018) and for temporal networks (Hulovatyy et al., 2015). In the first section of the results, we observe that for node and edge homophily measures, the most homophilic network representations are $\text{LINE}_{G_k}$ and $\text{A}_{G_k}$, while for GSI the most homophilic network representations are $\text{DeepWalk}_{G_k}$. This disagreement highlights the lack of a single universal measure of homophily that could act as a proxy for selecting the most suitable ML method (linear or nonlinear) for downstream analysis tasks, that would ease

the computational burden. This is expected, since we are dealing with computationally intractable (NP-hard) problems, so it is the structure of the data that has to be exploited to construct mining algorithms that are efficient for the data of that particular structure. From the theory of computation we know that this is the best we can do for computationally intractable problems. Our study is the first to provide fundamental insights into the structural characteristics of network data that enable their **linear** mining and exploitation, hence providing the building blocks that enable the ML community to build upon them to efficiently and explainably mine complex network data.

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

# A  Appendix

## A.1  GDV similarity of the nodes

The state-of-the-art methods to precisely capture network wiring are based on graphlets: small, connected, non-isomorphic, induced sub-graphs of a large network that appear at any frequency (Pržulj et al., 2004). Different topological positions within graphlets are characterized by different symmetry groups of nodes, called automorphism orbits (Pržulj, 2007). Orbits describe the different topological roles of a node in a particular graphlet (see Figure 3). Yaveroğlu et al. (2014) showed that between the orbits, there exist redundancies, as well as dependencies, and proposed a set of 11 non-redundant orbits of 2- to 4-node graphlets (see Figure 3). Each node in the network is represented by its 11-dimensional vector called *Graphlet Degree Vector (GDV)*, that captures the 11 non-redundant graphlet degrees of the node. To quantify the topological similarity between two nodes, $u$ and $v$, we compare their GDV vectors using the GDV distance which is computed as follows. Given two GDV vectors, $x$ and $y$, the distance between their $i^{th}$ coordinate is defined as:

$$Dist_i(x,y) = w_i \times \frac{\log(x_i + 1) - \log(y_i + 1)}{\log(max\{x_i, y_i\} + 2)}, \tag{5}$$

where $w_i$ is the weight of orbit $i$ that accounts for dependencies between the orbits (see details in Milenković & Pržulj (2008a)). The log-scale is used to control the different orders of magnitude between orbit counts.

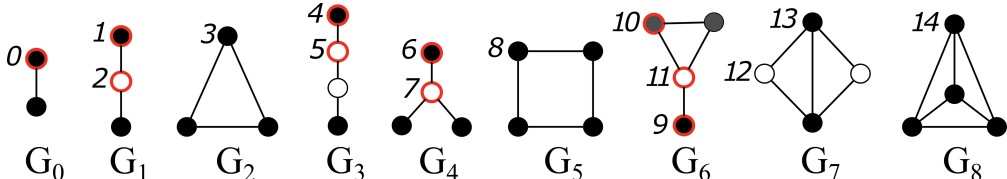

Figure 3: **The nine 2- to 4-node graphlets and their 15 orbits.** Within each graphlet, $G_i, i \in \{0, \ldots, 8\}$, nodes belonging to the same orbit are of the same shade and are numbered from 0 to 14. The eleven non-redundant orbits, whose counts cannot be derived from the counts of the other orbits, are highlighted in red.

Then, GDV similarity is defined as:

$$GDVsim(u, v) = 1 - \frac{\sum_{i=1}^{11} Dist_i(x, y)}{\sum_{i=1}^{11} w_i}. \tag{6}$$

## A.2 Non-negative matrix tri-factorization based embeddings

In this study, we use an Orthonormal NMTF (ONMTF) framework to decompose the different representation of the networks ($A_{G_k}$, LINE$_{G_k}$, DeepWalk$_{G_k}$ and the pure graphlet-based representations GDV PPMI and GDV sim matrix) to generate the corresponding network embedding spaces. To define the dimensionality of the embedding space, $d$, we use the heuristic rule of thumb, $d = \sqrt{\frac{n}{2}}$, where $n$ is the number of nodes in the network (Kodinariya & Makwana, 2013).

In particular, given an input network matrix representation, $X$, the ONMTF framework decomposes it into three non-negative matrix factors, $E$, $S$ and $P^T$, as $X \approx ESP^T$, where $E \cdot S$ contains the vector representations (embeddings) of the entities of $X$ in the embedding space spanned by $P^T$, $S$ is a compressed representation of network $X$ and $P^T$ is the orthonormal basis of the embedding space. Importantly, the orthonormality constraint ($P^T P = I$) leads to independent, nonambiguous directions in the embedding space and improves the clustering interpretation of NMTF (Ding et al., 2006). The decomposition is done by solving the following:

$$min_{G,S,P \geq 0} \|X - ESP^T\|_F^2, P^T P = I \tag{7}$$

where F denotes the Frobenius Norm. This optimization problem is NP-hard (Ding et al., 2006), thus to solve it we use a fixed point method that starts from an initial solution and iteratively uses the multiplicative update rules (Ding et al., 2006), derived from the Karush-Kuhn-Tucker (KKT) conditions, to converge towards a locally optimal solution (see Xenos et al. (2021) for details). To generate the initial $E$, $S$ and $P$ matrices, we use the Singular Value Decomposition (SVD) based strategy (Qiao, 2015). This strategy makes the solver deterministic and also reduces the number of iterations that are needed to achieve convergence (see Qiao (2015) for more details). We stop the iterative solver after 500 iterations.

## A.3 Biological data collection and network construction

In this study, we focus on the most well-studied molecular networks, modelling protein-protein interactions and gene co-expressions of the following three species: *Homo sapiens* (human), *Saccharomyces cerevisiae* (budding yeast) and *Schizosaccharomyces pombe* (fission yeast). For each of these three organisms, we collected the experimentally validated protein-protein interactions (PPIs) from BioGRID (Oughtred et al., 2021) version 3.5.182. We model these data as PPI networks in which nodes represent protein-coding genes, and edges connect nodes (genes) whose protein products physically bind. Also, we collected the gene co-expressions (COEX) from COXPRESdb Obayashi et al. (2022) version 8 (for all the species, we selected the dataset, u22, that has the highest number of expression samples). In our collected co-expression datasets, the co-expression measure between two genes is standardized using zeta scores. To construct the COEX networks, in which nodes represent genes and edges the co-expressions between the genes, we selected the strongest co-expression values having a zeta score higher than or equal to 3, which is the usual practice. The

Table 4: **Statistics of the multi-labeled biological networks.** For each species (row), the table shows the corresponding number of nodes, edges, and density for the PPI (columns 2, 3, and 4) and COEX (columns 5, 6, and 7) networks.

| Species | PPI | | | COEX | | |
|---|---|---|---|---|---|---|
| | #Node | #Edges | Density | #Node | #Edges | Density |
| *Homo sapiens* (Human) | 18,614 | 398,713 | 0.0023 | 16,808 | 1,610,160 | 0.0114 |
| *Saccharomyces cerevisiae* (Baker's yeast) | 5,886 | 110,451 | 0.0064 | 5,647 | 167,560 | 0.0105 |
| *Schizosaccharomyces pombe* (Fission yeast) | 3,214 | 10,923 | 0.0021 | 4,951 | 504,790 | 0.0412 |

statistics of the six generated biological networks are presented in Appendix Table 4. These networks are multi-labeled, as a gene often has multiple annotations, detailed below.

**Gene labels (annotations):** For each gene (or equivalently, protein, as a gene product) in the biological networks, we collected their Reactome Pathway (RP) annotations from Reactome database v83 (Fabregat et al., 2017) and their KEGG Pathway (KP) annotations from KEGG database v111.1 (Kanehisa & Goto, 2000). Also, we collected the most specific experimentally validated Gene Ontology Biological Process (GO BP) annotations (The Gene Ontology Consortium, 2025) from the NCBI database (downloaded on September 2023). The GO BP annotations are back-propagated, using the *is-a* relationship, to their ancestors in the Gene Ontology tree. Note that a gene can have multiple GO BP, KEGG and RP annotations. The number of annotated genes for each type of annotations are presented in the Supplementary Tables: 6, 8 and 9 (presented in Appendix A.7).

## A.4   Random partition graph model

To demonstrate the relationship between the homophily level of the input network matrix representation and the linear separability in the embedding space under ideal conditions, without noisy data, we simulate data using the random partition graph model (Fortunato, 2010). The main parameters of this model are the number of partitions (representing communities), the size of each community, the probability of connections between nodes within the same community ($p_{in}$, intra-edges), and the probability of connections between nodes from different communities ($p_{out}$, inter-edges). We simulate 420 networks, each comprising 1000 nodes distributed across 5 communities, by varying the two probabilities, $p_{out}$ and $p_{in}$, from 0 to 1 in increments of 0.05. For each simulated network, we compute the homophily level of the input network matrix representation: adjacency matrix, DeepWalk closed matrix formula and LINE closed matrix formula. Then, we use our NMTF-based framework to generate the embedding space, and for each embedding space, we perform node classification using SVM with the Euclidean kernel. Finally, we compute the Pearson Correlation Coefficient between the homophily levels of the input network matrix representations and the F1-node classification scores.

## A.5   Downstream analysis tasks

To demonstrate that higher linear separability of classes in the embedding space leads to better downstream analysis results, we perform two different experiments: one for single-labeled networks and another for multi-labeled biological networks. For the single-labeled networks, we predict the label of the nodes based on their distances in the embedding space. For the biological networks, where nodes (genes) can have multiple annotations or remain unannotated due to unknown functions, a common downstream analysis task involves uncovering functional network modules—sets of genes that collectively perform higher-level biological functions.

**Label prediction in single label networks:** in a single-label network, each node is assigned one label, and two nodes that have the same label belong to the same class. A common downstream task in an embedding

space is to classify a node based on its cosine similarity from nodes with known label. The cosine similarity is a measure of similarity between two vectors that considers the angle between them rather than their magnitude. Here, we evaluate if the embedding spaces of single-label networks are organized coherently with respect to their classes. To do so, we formulate the problem as a multi-class classification problem, in which we aim to classify two nodes with the same label based on their cosine similarity. The ability of the cosine similarity between the embedding vectors to correctly identify nodes coming from the same class is formally evaluated using ROC curve (AUROC) analysis (Bradley, 1997). Since we have more than one class (labels), we use the one-vs-rest strategy to generalize the binary classification task to the multi-class problem. In this strategy, we use one binary classifier for each possible class and compute its corresponding AUROC score. Then, to account for the imbalances between the different classes, we calculate the weighted AUROC score.

**Functional module discovery in biological networks:** in biological networks, a common downstream analysis task is uncovering functional network modules, i.e., sets of genes that together perform higher-level biological functions. After embedding a biological network, this is typically done by clustering the genes based on the proximity of their embedding vectors (Euclidean distance) in the embedding space. Subsequently, the biological relevance of the obtained gene clusters is measured by assessing if genes that cluster together significantly share biological annotations through over-representation (enrichment) analysis.

To assess whether genes that are close in the embedding space of the PPI and COEX networks perform higher-level biological functions, we cluster the genes using $k$-means clustering on the embedding vectors obtained from the orthogonal space, $E.S$, of the NMTF. The number of clusters, $k$, is determined using the heuristic rule of thumb, $k = \sqrt{\frac{n}{2}}$, where $n$ is the number of data points (Kodinariya & Makwana, 2013). We then evaluate the biological relevance of the obtained gene clusters by measuring the percentages of clusters that are enriched in Reactome Pathways (RP) terms. The probability that an annotation is enriched in a cluster is computed using the hypergeometric test (sampling without replacement strategy):

$$p = 1 - \sum_{i=0}^{X-i} \binom{K}{i}\binom{M-K}{N-i}/\binom{M}{N}, \tag{8}$$

where $N$ is the number of annotated genes in the cluster, $X$ is the number of genes in the cluster that are annotated with the given annotation, $M$ is the number of annotated genes in the network and $K$ is the number of genes in the network that are annotated with the annotation in question. An annotation is considered to be statistically significantly enriched if its enrichment $p$-value, after Benjamini and Hochberg correction (Benjamini & Hochberg, 1995) for multiple hypothesis testing, is lower than or equal to 5%.

We measure the functional coherence of the clustering by computing the percentage of genes (we term it "gene coverage") that are grouped together and share functions, i.e., that have at least one of their annotations enriched in their clusters over all annotated genes. In addition, we compute if the space captures all the possible functions of the genes, i.e., the percentage of annotations that are enriched (we term it "functional coverage") in at least one of the gene clusters.

### A.6 Appendix Figures and Tables

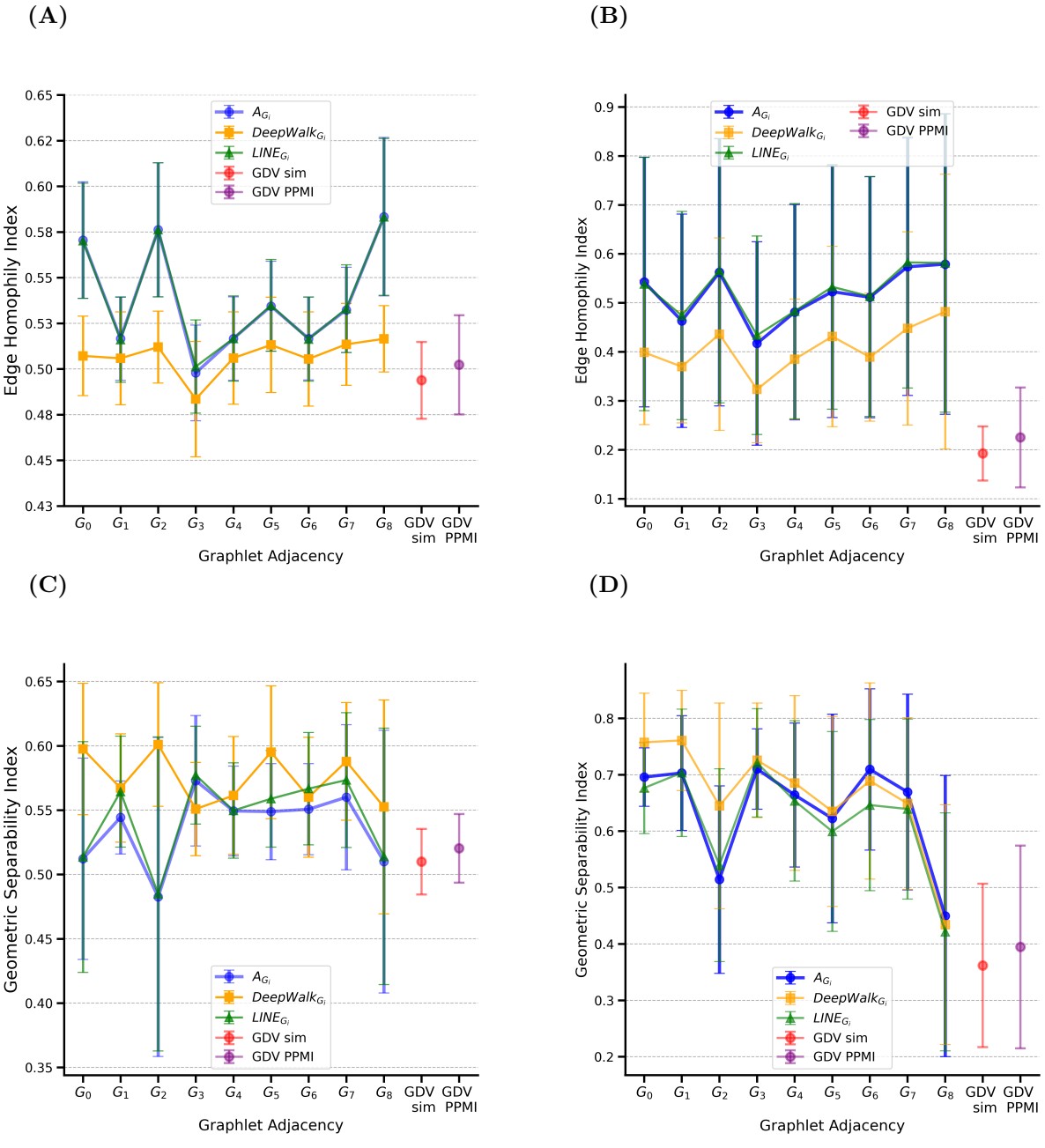

Figure 4: **Graphlet-based network matrix representations lead to more hopophilic representations.** In Panel **A**, for each graphlet (x-axis) and method (color-coded), the line plot shows the average, over the six biological multi-labeled networks, edge homophily index and the standard deviation. Panel **B** shows the same, but on average over the seven single-labeled networks. In panel **C**, for each graphlet (x-axis) and method (color-coded) the line plot shows the average Geometric Separability Index (y-axis), over the six biological multi-labeled networks, along with the standard deviation. Panel **D** shows the same, but on average over the seven single-labeled networks.

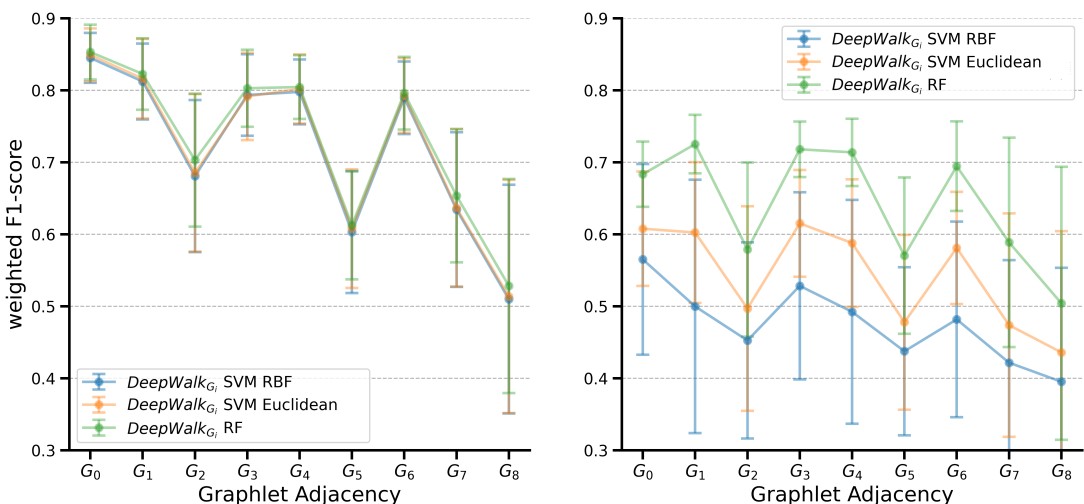

Figure 5: **DeepGraphlets node classification performances in the single-labeled networks.** In the left panel, for each of the nine graphlets (x-axis) and for each classifier (color-coded), the line plot shows for the corresponding nine DeepGraphlets based embeddings the weighted node classification F1-score averaged over the three fully linear single-labeled networks (Cora, CS Co-author and Wikipedia CS) and the standard deviation. The right panel shows the same, but for the four non-linearly separable single-labeled networks (Cameleon, Squirrel, CiteSeer and USA air-traffic).

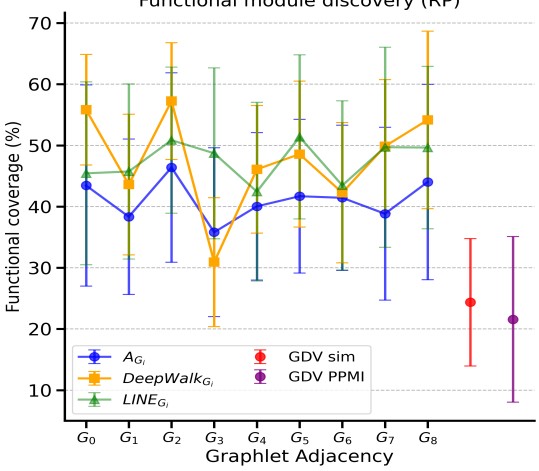

Figure 6: **Graphlet-based embeddings lead to better functional module discovery in gene embedding spaces for Reactome Pathway (RP) annotations.** For each graphlet (x-axis) and for each method (color-coded), the line plot shows on average (over the six multi-labeled biological networks) percentage of the enriched Reactome Pathway (RP) annotations (y-axis) and the standard deviation.

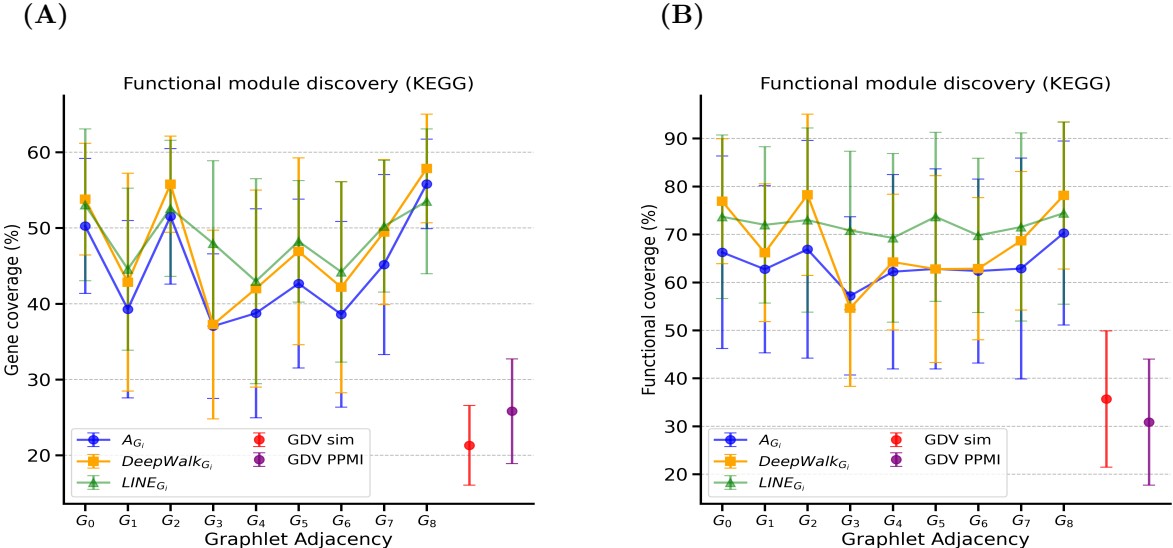

Figure 7: **Graphlet-based embeddings lead to better results in downstream analysis tasks for KEGG Pathway annotations.** Panel **A** presents the results of the functional module discovery in gene embedding spaces of the biological multi-labeled networks. In particular, for each graphlet (x-axis) and for each method (color-coded) the line plot shows the average percentage, over the six molecular networks, of annotated genes in the clusters that have at least one KEGG Pathway term enriched in their clusters, along with the standard deviation (y-axis). Panel **B** shows on average (over the six multi-labeled biological networks) percentage of the enriched KEGG Pathway annotations (y-axis) and the standard deviation.

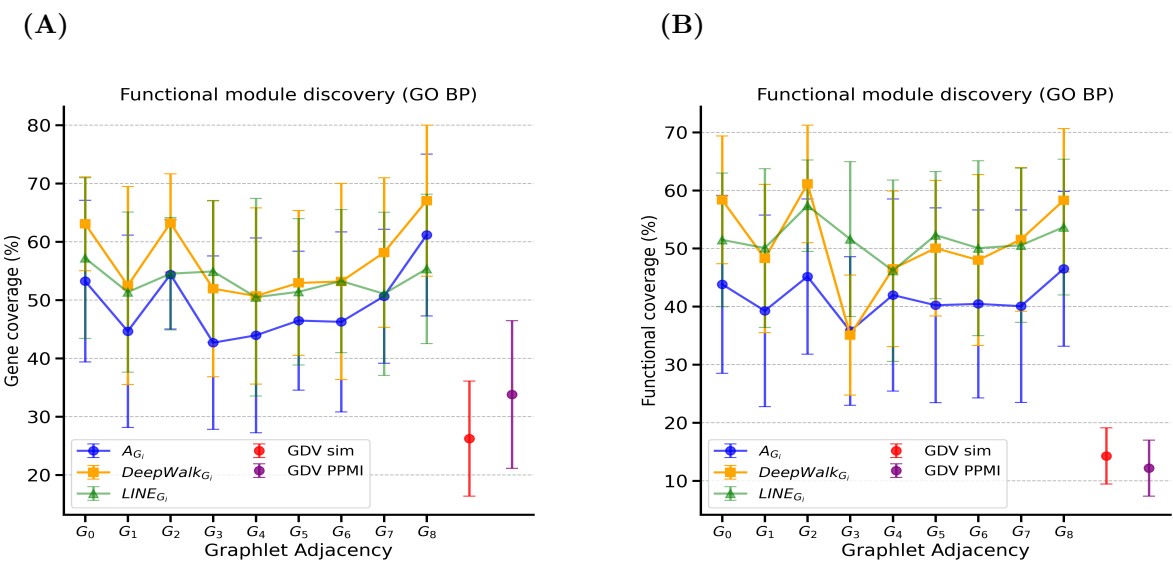

Figure 8: **Graphlet-based embeddings lead to better results in downstream analysis tasks for GO BP annotations.** Panel **A** presents the results of the functional module discovery in gene embedding spaces of the biological multi-labeled networks. In particular, for each graphlet (x-axis) and for each method (color-coded) the line plot shows the average percentage, over the six molecular networks, of annotated genes in the clusters that have at least one GO BP term enriched in their clusters, along with the standard deviation (y-axis). Panel **B** shows on average (over the six multi-labeled biological networks) percentage of the enriched GO BP annotations (y-axis) and the standard deviation.

Table 5: **Statistics of the single-labeled networks.** For each network (row), the table shows the corresponding number of labels (column 2), nodes (column 3), edges (column 4) and density (column 5). In addition, it shows the node homophily index (column 6) and the edge homophily index (column 7) computed on their adjacency matrix representations.

| Network | Labels (categories) | #Nodes | #Edges | Density | Node Homophily | Edge Homophily |
|---|---|---|---|---|---|---|
| Wikipedia CS | 10 | 10,657 | 145,133 | 0.0025 | 0.695 | 0.684 |
| Chameleon | 5 | 2,277 | 31,371 | 0.0121 | 0.247 | 0.229 |
| Squirrel | 5 | 5,201 | 188,239 | 0.0139 | 0.217 | 0.222 |
| Cora | 7 | 2,708 | 5,278 | 0.0014 | 0.825 | 0.809 |
| CiteSeer | 6 | 2,120 | 4,732 | 0.0012 | 0.711 | 0.735 |
| USA air-traffic | 4 | 1,186 | 13,597 | 0.0193 | 0.371 | 0.667 |
| CS Co-author | 15 | 18,333 | 81,894 | 0.0005 | 0.832 | 0.684 |

Table 6: **Numbers of annotated nodes in the biological multi-labeled networks.** For each species (row), the table shows the numbers of level 1 GO BP terms that annotate the nodes (column 2), the numbers of nodes in the PPI networks annotated with GO BP terms and the total numbers of nodes in the PPI networks (columns 3 and 4), and the numbers of nodes in the COEX networks annotated with GO BP terms and the total numbers of nodes in the COEX networks (columns 5 and 6).

| | GO BP Categories | PPI #Annotated Nodes | #Nodes | COEX #Annotated Nodes | #Nodes |
|---|---|---|---|---|---|
| *Homo sapiens* (Human) | 18 | 8,772 | 18,614 | 7,775 | 16,808 |
| *Saccharomyces cerevisiae* (Baker's yeast) | 12 | 4,591 | 5,886 | 4,517 | 5,647 |
| *Schizosaccharomyces pombe* (Fission yeast) | 11 | 1,413 | 3,214 | 1,669 | 4,951 |

Table 7: **Embedding space that corresponds to the maximum node classification F1-scores in the single-labeled and multi-labeled networks.** For each network (row), the table shows the graphlet-based embedding space in which the maximum weighted node classification F1-score of the corresponding classifier (linear SVM (L-SVM), SVM RBF, and RF) is achieved. The first 7 networks correspond to the single-labeled networks, and the last 6 networks correspond to the multi-labeled molecular networks. Bold cells denote the graphlet that yielded the highest value per row.

| Networks | DeepWalk$_{G_k}$ | | | LINE$_{G_k}$ | | | Graphlet Adjacency, A$_{G_k}$ | | |
|---|---|---|---|---|---|---|---|---|---|
| | L-SVM | SVM RBF | RF | L-SVM | SVM RBF | RF | L-SVM | SVM RBF | RF |
| Cora | $G_0$ | $G_0$ | $\mathbf{G_0}$ | $G_3$ | $G_3$ | $G_1$ | $G_3$ | $G_3$ | $G_3$ |
| Wikipedia CS | $\mathbf{G_0}$ | $G_0$ | $G_0$ | $G_3$ | $G_3$ | $G_0$ | $G_4$ | $G_4$ | $G_0$ |
| CS Co-author | $G_0$ | $G_0$ | $\mathbf{G_0}$ | $G_3$ | $G_3$ | $G_1$ | $G_3$ | $G_3$ | $G_0$ |
| USA air-traffic | $G_1$ | $G_6$ | $G_6$ | $G_7$ | $G_2$ | $\mathbf{G_2}$ | $G_6$ | $G_1$ | $G_0$ |
| Chameleon | $G_0$ | $G_0$ | $\mathbf{G_4}$ | $G_1$ | $G_1$ | $G_4$ | $G_6$ | $G_6$ | $G_1$ |
| CiteSeer | $G_3$ | $G_1$ | $\mathbf{G_1}$ | $G_3$ | $G_3$ | $G_1$ | $G_3$ | $G_3$ | $G_3$ |
| Squirrel | $G_3$ | $G_3$ | $G_4$ | $G_4$ | $G_1$ | $G_4$ | $G_1$ | $G_1$ | $\mathbf{G_1}$ |
| Pombe COEX | $G_2$ | $G_4$ | $G_0$ | $G_7$ | $\mathbf{G_3}$ | $G_3$ | $G_2$ | $G_4$ | $G_7$ |
| Cerevisiae COEX | $\mathbf{G_0}$ | $G_0$ | $G_0$ | $G_3$ | $G_6$ | $G_3$ | $G_6$ | $G_2$ | $G_5$ |
| Homo sapiens COEX | $G_8$ | $\mathbf{G_0}$ | $G_2$ | $G_2$ | $G_3$ | $G_7$ | $G_0$ | $G_1$ | $G_7$ |
| Pombe PPI | $\mathbf{G_0}$ | $G_0$ | $G_0$ | $G_2$ | $G_3$ | $G_7$ | $G_2$ | $G_3$ | $G_2$ |
| Cerevisiae PPI | $\mathbf{G_8}$ | $G_8$ | $G_8$ | $G_8$ | $G_8$ | $G_8$ | $G_8$ | $G_8$ | $G_8$ |
| Homo sapiens PPI | $\mathbf{G_2}$ | $G_2$ | $G_8$ | $G_8$ | $G_2$ | $G_8$ | $G_8$ | $G_0$ | $G_8$ |

Table 8: **Numbers of annotated nodes in the biological multi-labeled networks with KEGG pathway terms.** For each species (row), the table shows the numbers of nodes in the PPI networks annotated with KEGG pathway terms and the total numbers of nodes in the PPI networks (columns 2 and 3), and the numbers of nodes in the COEX networks annotated with KEGG terms and the total numbers of nodes in the COEX networks (columns 4 and 5).

| | PPI | | COEX | |
|---|---|---|---|---|
| | #Annotated Nodes | #Nodes | #Annotated Nodes | #Nodes |
| *Homo sapiens* (Human) | 7,770 | 18,614 | 6,836 | 16,808 |
| *Saccharomyces cerevisiae* (Baker's yeast) | 2,131 | 5,886 | 2,109 | 5,647 |
| *Schizosaccharomyces pombe* (Fission yeast) | 1,298 | 3,214 | 1,712 | 4,951 |

Table 9: **Numbers of annotated nodes in the biological multi-labeled networks with Reactome Pathway terms.** For each species (row), the table shows the numbers of nodes in the PPI networks annotated with RP terms and the total numbers of nodes in the PPI networks (columns 2 and 3), and the numbers of nodes in the COEX networks annotated with RP terms and the total numbers of nodes in the COEX networks (columns 4 and 5).

| | PPI | | COEX | |
|---|---|---|---|---|
| | #Annotated Nodes | #Nodes | #Annotated Nodes | #Nodes |
| *Homo sapiens* (Human) | 9,951 | 18,614 | 8,934 | 16,808 |
| *Saccharomyces cerevisiae* (Baker's yeast) | 1,699 | 5,886 | 1,688 | 5,647 |
| *Schizosaccharomyces pombe* (Fission yeast) | 1,163 | 3,214 | 1,449 | 4,951 |

