# OpenReview forum: "Simplifying complex machine learning by linearly separable network embedding spaces"
_TMLR — Under review for TMLR_

### Review · Reviewer_bu8G · 2026-04-12

**Summary Of Contributions:**

**Summary.** The paper proposes graphlet-based extensions of LINE and DeepWalk (LINEGk, DeepWalkGk) that produce more homophilic network matrix representations. The central claim is that more homophilic representations yield more linearly separable embedding spaces, which in turn enable simpler (linear) downstream classifiers to perform comparably to nonlinear ones. This is validated on 13 real-world networks and synthetic data.

## Strengths

The core insight, linking homophily of the input representation to linear separability of the embedding space, is clean and potentially useful. If you can transform a heterophilic network into a more homophilic representation *without* using labels, you sidestep the need for complex GNN architectures. That's a compelling narrative, and the paper delivers on it structurally: the logical chain (graphlet representations -> more homophily -> more linear separability -> better downstream results) is laid out clearly across Sections 4.2–4.6.

The breadth of experiments is reasonable: six biological multi-label networks, seven single-label benchmarks, synthetic data from the random partition model, three classifier types, multiple homophily measures. The graphlet-based extensions themselves are natural and well-motivated from the network biology literature.

The connection to the linear representation hypothesis in NLP/VLMs (Park et al., 2024) is a nice framing that positions the work within a broader trend.

## Weaknesses and Concerns

1. **The correlations are moderate and the causal claim is overstated.** The Pearson correlations between homophily and linear SVM F1-scores range from 0.19 to 0.68 (Table 2). The node/edge homophily correlations on real data hover around 0.35–0.53. These are statistically significant but not strong enough to support statements like "we show that the more homophilic the network representation, the more linearly separable the corresponding network embedding space." The relationship is noisy, and the paper doesn't adequately discuss what other factors (density, community structure, label distribution) might confound this correlation.

2. **The "linearly separable" threshold is arbitrary and generous.** Defining "fully linearly separable" as linear SVM F1 ≥ 0.8, and "sufficiently linearly separable" as linear SVM ≥ RBF SVM (regardless of absolute performance), is quite loose. An embedding space where linear SVM gets 0.50 and RBF SVM gets 0.49 would be called "sufficiently linearly separable," which seems misleading. The paper would benefit from a more principled or continuous measure of linear separability rather than these ad hoc bins.

3. **No comparison with modern GNN baselines.** The paper positions itself against computationally intensive ML systems but only compares against SVM, RF, and the factorization-based embedding pipeline. There's no comparison with GCN, GAT, GraphSAGE, or any heterophily-specialized GNN (e.g., H2GCN, LINKX). The claim that linear methods can replace complex ML is hard to evaluate without showing what the complex ML actually achieves on these datasets. The Chameleon/Squirrel results (F1 around 0.5–0.77) are well below what modern methods achieve.

4. **The graphlet approach uses no labels, but the homophily measurement does.** The paper emphasizes that the graphlet-based representations are purely structural and don't require labels. But the entire evaluation pipeline (measuring homophily, defining linear separability, node classification) requires labels. So the practical workflow is unclear: how does a practitioner know *which* graphlet representation to use without labels? The paper doesn't address this model selection problem.

5. **Scalability is not discussed.** Computing graphlet adjacency matrices for all nine 2-to-4-node graphlets on networks with 18K+ nodes is expensive. The paper doesn't report runtimes or discuss computational cost, which undercuts the "simplifying complex ML" narrative. If the preprocessing is itself computationally intensive, the savings from using a linear classifier downstream may be negligible.

**Audience:**

Yes

**Audience Explanation:**

The paper's core insight linking homophily to linear separability is relevant to researchers working on GNNs for heterophilic graphs, network embedding methods, and computational biology. The graphlet-based extensions of LINE and DeepWalk are also of practical interest to the network mining community.

**Claims And Evidence:**

No

**Claims Explanation:**

The central claim that more homophilic representations yield more linearly separable embeddings is supported only by moderate Pearson correlations (0.19–0.68, with real-data values around 0.35–0.53). While statistically significant, these do not convincingly support the causal language used, and confounding factors (density, community structure, label distribution) are not discussed.

The definition of "linear separability" is arbitrary: an embedding where linear SVM scores 0.50 and RBF SVM scores 0.49 would be deemed "sufficiently linearly separable," inflating support for the claims.

No comparison with modern GNN baselines (GCN, GAT, LINKX, H2GCN) is provided. Without showing what the "computationally intensive" methods actually achieve, the claim that linear classifiers can replace them is unsupported. The Chameleon/Squirrel F1 scores (0.5–0.77) are well below modern benchmarks.

The graphlet representations are claimed to be label-free, yet selecting *which* graphlet to use requires labels in practice. This model selection gap is not addressed. No runtime analysis is provided either, leaving the "simplifying complex ML" narrative unconvincing when the preprocessing itself may be expensive.

**Requested Changes:**

See summary.

---

> ### Author Response · Authors · 2026-07-03
> **Rebuttal Concern 1 Part 1**
>
> We thank the reviewer for the constructive feedback and for recognising the core insight of our work: linking representation homophily to linear separability. Below, we address each point, and we will incorporate the corresponding new experiments into the revised manuscript.
>
> **Reviewer Comment 1:**
> "The correlations are moderate and the causal claim is overstated."
>
> **Our response:**
>
> We thank the reviewer for the comment. We agree that the original phrasing overstated the relationship, and we respond in two ways: (1) we soften the claim, and (2) we show that the moderate pooled correlation is largely driven by a small **representation coverage** (fraction of nodes in the largest connected component of a representation) in a subset of graphlet-based representations. When excluding the matrix representations having smallest coverage (<= 50% of the nodes), our observed correlations become much stronger and are not affected by any of the tested confounders (density, community structure, clustering coefficient and label distribution).
>
> **1. Softened, associational language.** We did not intend a causal claim. We revise all such statements throughout — e.g. "the more homophilic the network representation, the more linearly separable the corresponding embedding space" becomes "homophily is positively associated with linear separability."
>
> **2. Why the pooled correlation looks moderate.**  We hypothesise that the moderate averaged accross network type correlations (0.35–0.53 on the real networks) are largely an artefact of a subset of incomplete representations, i.e., representations with small coverage. The two citation networks are the clearest case — on Cora and CiteSeer, coverage collapses from ~1.0 for small *k* to ~0.03 at *k*=8 (Table 1, bold cells). On such fragmented representations homophily is computed over the few surviving same-class edges and is artificially inflated while F1 collapses. Including these incomplete representations in the pooled average is exactly what drags the correlation into the "moderate" range. The same incomplete views occurs for the Pombe PPI network for $k{=}4$ and $k{=}8$ (Table 2).
>
> *Table 1 — Representation coverage, single-label networks. Fraction of nodes in the largest connected component of the graphlet adjacency, per graphlet adjacency G0–G8 (identical across the three embedding families). **Bold** cells are below 0.5 representation coverage and are excluded from the filtered correlations.*
>
> | Network | G0 | G1 | G2 | G3 | G4 | G5 | G6 | G7 | G8 |
> |---|---|---|---|---|---|---|---|---|---|
> | Cora | 0.92 | 0.92 | **0.34** | 0.92 | 0.87 | **0.32** | 0.81 | **0.25** | **0.03** |
> | CiteSeer | 1.00 | 1.00 | **0.13** | 1.00 | 0.85 | **0.17** | 0.65 | **0.12** | **0.03** |
> | Wikipedia CS | 1.00 | 1.00 | 0.80 | 1.00 | 0.99 | 0.79 | 0.99 | 0.79 | 0.61 |
> | CS Co-author | 1.00 | 1.00 | 0.85 | 1.00 | 0.98 | 0.59 | 0.98 | 0.77 | 0.59 |
> | USA air-traffic | 1.00 | 1.00 | 0.80 | 1.00 | 1.00 | 0.64 | 0.99 | 0.79 | 0.62 |
> | Chameleon-filt. | 1.00 | 1.00 | 0.87 | 1.00 | 0.97 | 0.68 | 0.99 | 0.81 | 0.56 |
> | Squirrel-filt. | 1.00 | 1.00 | 0.89 | 1.00 | 0.99 | 0.84 | 0.99 | 0.87 | 0.71 |
>
> *Table 2 — Representation coverage, biological multi-label networks. **Bold** cells are below 0.5 representation coverage and are excluded from the filtered correlations.*
>
> | Network | G0 | G1 | G2 | G3 | G4 | G5 | G6 | G7 | G8 |
> |---|---|---|---|---|---|---|---|---|---|
> | PPI (pombe) | 1.00 | 1.00 | **0.49** | 1.00 | 0.99 | 0.55 | 0.97 | **0.47** | **0.25** |
> | Co-exp (pombe) | 1.00 | 1.00 | 0.95 | 1.00 | 0.99 | 0.94 | 0.99 | 0.95 | 0.90 |
> | PPI (yeast) | 1.00 | 1.00 | 0.92 | 1.00 | 1.00 | 0.93 | 1.00 | 0.92 | 0.83 |
> | Co-exp (yeast) | 1.00 | 1.00 | 0.98 | 1.00 | 1.00 | 0.97 | 1.00 | 0.98 | 0.94 |
> | PPI (human) | 1.00 | 1.00 | 0.77 | 1.00 | 1.00 | 0.86 | 1.00 | 0.77 | 0.56 |
> | Co-exp (human) | 1.00 | 1.00 | 1.00 | 1.00 | 1.00 | 1.00 | 1.00 | 1.00 | 1.00 |
>
> In the following tables, we assess the coverage hypothesis and the confounders raised by the reviewer: density; community structure (average clustering and Louvain modularity); and label distribution.
>
> **(a) Coverage filtering strengthens the homophily–separability association** and, at the same time, removes spurious confounder correlations. On the single-label networks, node-homophily rises from +0.08 (all 189 representations) to +0.56/+0.58 (coverage > 0.5 / > 0.8) for the linear SVM. Average clustering (+0.41) and modularity (−0.31) carry non-trivial *unfiltered* correlations that **collapse to ≈0** once incomplete representations are excluded, while density stays near zero throughout — i.e. filtering cleans an artefact common to all descriptors.

---

> > ### Author Response · Authors · 2026-07-03
> > **Rebuttal Concern 1 Part 2**
> >
> > *Table 3 — single-label networks: Pearson correlation of each measure with node-classification F1 (L-SVM / shrinkage-LDA), unfiltered and under the coverage filters.*
> >
> > | Measure | Unf. (L-SVM) | >0.5 (L-SVM) | >0.5 (LDA) | >0.8 (L-SVM) | >0.8 (LDA) |
> > |---|---|---|---|---|---|
> > | Node homophily | +0.08 | +0.56 | +0.47 | +0.58 | +0.50 |
> > | Edge homophily | +0.04 | +0.52 | +0.42 | +0.53 | +0.44 |
> > | Adjusted homophily | +0.08 | +0.54 | +0.45 | +0.57 | +0.49 |
> > | Label informativeness | +0.01 | +0.50 | +0.38 | +0.55 | +0.44 |
> > | GSI | +0.71 | +0.84 | +0.83 | +0.94 | +0.93 |
> > | Density | +0.03 | −0.18 | −0.12 | −0.22 | −0.15 |
> > | Avg. clustering | +0.41 | −0.07 | −0.09 | −0.05 | −0.06 |
> > | Modularity (Louvain) | −0.31 | +0.09 | +0.11 | +0.24 | +0.24 |
> >
> > The biological networks show the same pattern: node homophily rises from +0.49 (L-SVM) to +0.72 at coverage > 0.5, while density, clustering and modularity stay near zero.
> >
> > *Table 4 — biological multi-label networks. Pearson correlations. "All" = every representation (162); "Cov > 0.5" (153) drops fragmented adjacencies. (Adjusted homophily / label informativeness are single-label constructs and omitted.)*
> >
> >
> > | Measure | All (L-SVM) | All (LDA) | >0.5 (L-SVM) | >0.5 (LDA) |
> > |---|---|---|---|---|
> > | Node homophily | +0.49 | +0.25 | +0.72 | +0.58 |
> > | Edge homophily | +0.43 | +0.33 | +0.57 | +0.54 |
> > | GSI | +0.42 | +0.21 | +0.48 | +0.35 |
> > | Density | −0.09 | +0.00 | −0.10 | −0.04 |
> > | Avg. clustering | +0.05 | +0.05 | +0.03 | −0.01 |
> > | Modularity (Louvain) | +0.03 | −0.09 | +0.06 | +0.00 |
> >
> > **(b) Label distribution.** This is addressed directly by two of the descriptors we already correlate: adjusted homophily and label informativeness (Platonov et al., 2023), which by design correct for the number of classes and class-size imbalance — i.e. for label distribution. Both track linear separability after correcting for the agreement expected at random (adjusted homophily ≈ +0.49–0.57; label informativeness ≈ +0.44–0.55; Table 3), so the association is not an artefact of label distribution. The noise-free networks simulated from the random-partition model (Table 5), which by design holds the label distribution fixed, confirm this independently: homophily stays positive (+0.20) while density and clustering remain close to zero.
> >
> > *Table 5 — simulated networks using the random-partition model: Pearson (P)correlation of each measure with node-classification F1 (L-SVM / shrinkage-LDA)
> > .*
> >
> > | Measure | P (L-SVM) | P (LDA) |
> > |---|---|---|
> > | Node homophily | +0.20 | +0.20 |
> > | Edge homophily | +0.20 | +0.20 |
> > | Adjusted homophily | +0.21 | +0.21 |
> > | Label informativeness | +0.25 | +0.24 |
> > | GSI | +0.68 | +0.68 |
> > | Density | −0.07 | −0.08 |
> > | Avg. clustering | +0.03 | +0.01 |
> > | Modularity (Louvain) | +0.29 | +0.29 |
> >
> > **Note on modularity in the Random-Partition Model.** Note on modularity in the random-partition model. Modularity also correlates positively here (+0.29; Table 5). This is expected, since the model generates graphs by first splitting the nodes into communities and then adding within- and between-community edges at fixed rates $p_\text{in}/p_\text{out}$, so modularity is built into the generator and is mechanically coupled to homophily (both are monotone in $p_\text{in}/p_\text{out}$); the model cannot separate them by construction. The real networks disentangle them (Tables 3–4): there modularity does not track separability (≤ +0.24 on single-label nets, near zero or negative on the biological nets), while homophily is consistently the stronger signal.
> >
> >
> > ## Conclusion
> > The moderate pooled correlation is largely an artefact of incomplete high-order graphlet representations, which inflate homophily on the few surviving edges while their classification F1 collapses. Once these incomplete representations are removed, the homophily–separability association is consistently positive and substantially stronger than the pooled average suggested. Moreover, none of the potential confounders shows a comparable association with separability on the real networks — density, average clustering, and Louvain modularity all remain near zero — and label distribution is excluded both by the chance-corrected homophily descriptors and by the label-fixed simulations. Homophily is therefore the structural property that most consistently associates with linear separability.

---

> ### Author Response · Authors · 2026-07-03
> **Rebuttal Concern 2**
>
> **Reviewer Comment 2:**
> "The 'linearly separable' threshold is arbitrary and generous."
>
> **Our response:**
>
> We thank the reviewer and agree that the original two-part definition and particularly the relative "linear SVM ≥ RBF SVM" criterion was too loose. We had introduced the relative criterion to be able to also characterize the multi-label biological networks, whose weighted F1 ≈ 0.5–0.66 is relatively high for the multi-label setting yet not comparable to a single-label F1-score.
>
> To simplify our results, we drop the relative criterion entirely. We now use a single, absolute definition, applied only to the single-label networks: an embedding space is linearly separable if its linear-SVM node-classification F1 is ≥ 0.8, and non-separable otherwise. Under this stricter criterion, the 0.50-vs-0.49 reviewer's example is correctly classified as non-separable. For the multi-label biological networks we do not apply this binary label, since a single-label F1 cutoff is not directly comparable to multi-label weighted F1.
>
> Second, and more importantly, our central finding does not depend on this threshold. The homophily–separability relationship is reported throughout as a continuous correlation between homophily and linear-SVM F1 (Tables 3–5), not as a count of networks above a cutoff. The ≥ 0.8 threshold is a conventional high-F1 bar, used only to sort the single-label networks into linear vs non-linear for discussion; we note explicitly that it is a convention, not a derived constant.
>
> The table below reports the linear-SVM F1 per network and embedding family and the corresponding best graphlet-based representation (k = 0, 1, …, 8) in parenthesis. Under the ≥ 0.8 criterion, three networks — Cora, Wikipedia CS, and CS Co-author — are linearly separable.
>
> *Linear-SVM node-classification F1 (mean±std; over the official splits for the single-label nets, over 10 seeds for the biological nets) for the three ONMTF graphlet-embedding families, the best graphlet-based extensions (k = 0, 1, …, 8) is in parenthesis. Under the ≥ 0.8 criterion, Cora, Wikipedia CS and CS Co-author are linearly separable.*
>
> | Network | DeepWalk_Gi | LINE_Gi | A_Gi |
> |---|---|---|---|
> | Cora | **0.827±0.017 (k=0)** | 0.792±0.010 (k=3) | 0.754±0.010 (k=3) |
> | CiteSeer | 0.704±0.027 (k=1) | 0.672±0.022 (k=1) | 0.663±0.020 (k=3) |
> | Wikipedia CS | **0.822±0.006 (k=0)** | 0.778±0.007 (k=3) | 0.740±0.004 (k=0) |
> | CS Co-author | **0.901±0.002 (k=0)** | **0.853±0.004 (k=3)** | **0.834±0.005 (k=3)** |
> | USA air-traffic | 0.636±0.023 (k=1) | 0.674±0.019 (k=7) | 0.663±0.026 (k=2) |
> | Chameleon-filt. | 0.388±0.047 (k=6) | 0.393±0.026 (k=0) | 0.367±0.029 (k=1) |
> | Squirrel-filt. | 0.359±0.019 (k=3) | 0.375±0.026 (k=0) | 0.337±0.013 (k=3) |
> | PPI (human) | 0.528±0.001 (k=2) | 0.524±0.001 (k=8) | 0.509±0.002 (k=8) |
> | PPI (yeast) | 0.647±0.002 (k=2) | 0.635±0.002 (k=0) | 0.606±0.001 (k=2) |
> | PPI (pombe) | 0.661±0.002 (k=0) | 0.588±0.002 (k=6) | 0.563±0.005 (k=2) |
> | Co-exp (human) | 0.515±0.001 (k=8) | 0.520±0.001 (k=1) | 0.517±0.001 (k=2) |
> | Co-exp (yeast) | 0.606±0.001 (k=2) | 0.610±0.001 (k=3) | 0.602±0.001 (k=6) |
> | Co-exp (pombe) | 0.527±0.003 (k=8) | 0.527±0.003 (k=7) | 0.530±0.002 (k=2) |
> | **Average** | 0.625 | 0.611 | 0.591 |

---

> ### Author Response · Authors · 2026-07-03
> **Rebuttal Concern 3 Part 1**
>
> **Reviewer Comment 3:**
> "No comparison with modern GNN baselines."
>
> **Our response:**
>
> We thank the reviewer for the suggestion and we include GCN and GAT. We also include Motif-GNN and HONE (Higher-Order Network Embedding) [Rossi et al., 2018] as additional baselines following Reviewer g9af's suggestion. Note that HONE is a structural baseline, which learns a single embedding jointly from all motif/graphlet adjacencies. Following Reviewer 6KPH, we now use the leakage-free filtered Chameleon (890 nodes) and Squirrel (2,223) [Platonov et al., 2023] and we also use two additional heterophilic benchmarks Amazon-ratings (24,492 nodes) and Roman-empire (22,662 nodes) [Platonov et al., 2023]; for training the GNNs we use the official node features and splits where available. Note that since the GNNs are trained to predict node class labels from both the graph structure and additional node features, are expected to perform better than purely graph structural approaches (which don’t use any node features to predict node labels), such as ours.
>
> We present the single-label and multi-label networks separately, since the latter have no node features.
>
> **1. Feature-rich single-label networks: GNNs lead, but narrowly.** Where informative node features exist, the GNNs are the best method on 7 of 9 networks (Table below). On the homophilic networks the margins are small (e.g. Cora 0.863 vs. our 0.840; WikiCS 0.783 vs. 0.765). Our purely structural embeddings still yield the best F1 on USA air-traffic and Amazon-ratings, which is a large heterophilic netowrk. The one clear case where GNNs outperform the structure-based embeddings is Roman-empire, a very sparse network (density = 0.00013) with essentially no structural signal (node homophily ≈ 0.046), so a structure-only representation is expected to be near-random. This is exactly what we observe (F1-scores in the range of 0.13 to 0.18), whereas the NN-based architectures that also use the node features yield F1-scores in the range of 0.53 to 0.61. On the small filtered heterophilic networks, all methods — GNNs included — score in the low 0.4s (Chameleon 0.40–0.43, Squirrel 0.33–0.43). The substantially higher numbers reported elsewhere are due to the label leakage in the unfiltered versions [Platonov et al., 2023].
>
> *Table — single-label networks, macro $F_1$​. Best non-NN embedding = best over the three ONMTF families, winning family in parentheses; bold = best per row.*
>
> | Network | Best non-NN embedding | HONE | GCN | GAT | Motif-GNN |
> |---|---|---|---|---|---|
> | Cora | 0.840 (DeepWalk_G0) | 0.797 | **0.863** | 0.862 | 0.859 |
> | CiteSeer | 0.721 (DeepWalk_G0) | 0.711 | 0.749 | 0.747 | **0.751** |
> | WikiCS | 0.765 (DeepWalk_G0) | 0.716 | **0.783** | 0.781 | 0.768 |
> | CoAuthor-CS | 0.901 (DeepWalk_G0) | 0.866 | **0.936** | 0.930 | 0.935 |
> | USA air-traffic | **0.684 (LINE_G2)** | **0.684** | 0.627 | 0.626 | 0.638 |
> | Chameleon-filt. | 0.406 (DeepWalk_G0) | 0.416 | **0.433** | 0.404 | 0.412 |
> | Squirrel-filt. | 0.391 (LINE_G0) | 0.379 | 0.379 | 0.332 | **0.432** |
> | Roman-empire | 0.179 (A_G0) | 0.113 | 0.527 | 0.592 | **0.606** |
> | Amazon-ratings | **0.517 (DeepWalk_G6)** | 0.510 | 0.386 | 0.399 | 0.230 |
>
> **2. Feature-free biological networks: structure wins.** The multi-label biological networks have no node features, so both our embeddings and the GNNs use only structural information here. The GNNs are given feature-free structural inputs: a learned node-identity input (results are comparable with $\log_2$-binned degree features). The GNNs are trained end-to-end with a per-class binary-cross-entropy objective and evaluated by weighted $F_1$ over 10 seeds. When features are not available, there is a graphlet-based embedding coupled with a simple linear classifier that outperform all three GNNs on every network. Notably, the structure-only HONE baseline also outperforms every GNN here while staying below our graphlet embeddings throughout. This shows that the advantage of structure-based embeddings on feature-free data is robust, and that our graphlet embeddings are the strongest among them.
>
> *Table — biological (multi-label) networks, weighted $F_1$ (mean ± std over 10 seeds). Best non-NN embedding = best over the three ONMTF families (best classifier in $\{$L-SVM, RBF-SVM, RF$\}$; winning family in parentheses); bold = best per row.*
>
> | Network | Best non-NN embedding | HONE | GCN | GAT | Motif-GNN |
> |---|---|---|---|---|---|
> | PPI (human) | **0.528 (DeepWalk$_{G_2}$)** | 0.513 | 0.449 | 0.452 | 0.456 |
> | PPI (yeast) | **0.647 (DeepWalk$_{G_2}$)** | 0.644 | 0.529 | 0.514 | 0.472 |
> | PPI (pombe) | **0.661 (DeepWalk$_{G_0}$)** | 0.611 | 0.525 | 0.552 | 0.434 |
> | Co-exp (human) | **0.524 (DeepWalk$_{G_8}$)** | 0.505 | 0.463 | 0.455 | 0.447 |
> | Co-exp (yeast) | **0.610 (LINE$_{G_3}$)** | 0.598 | 0.512 | 0.514 | 0.489 |
> | Co-exp (pombe) | **0.530 (A$_{G_2}$)** | 0.520 | 0.454 | 0.468 | 0.452 |

---

> > ### Author Response · Authors · 2026-07-03
> > **Rebuttal Concern 3 Part 2**
> >
> > The results of these comparisons place our work more precisely in the context of GNNs, and we revise the manuscript accordingly: when informative node features are available, GNNs perform better than the feature-free methods; but when features are not available — common in molecular and many other real-world networks — purely structural graphlet embeddings with a simple linear classifier match or outperform end-to-end GNNs, alleviating the need for them in this case.
> >
> > ## References
> >
> > - Rossi, R. A., Ahmed, N. K., Koh, E., Kim, S., Rao, A., & Abbasi-Yadkori, Y. (2018). HONE: Higher-Order Network Embeddings. *arXiv preprint* arXiv:1801.09303.
> > - Platonov, O., Kuznedelev, D., Diskin, M., Babenko, A., & Prokhorenkova, L. (2023). A critical look at the evaluation of GNNs under heterophily: Are we really making progress? *arXiv preprint* arXiv:2302.11640.

---

> ### Author Response · Authors · 2026-07-03
> **Rebuttal Concern 4**
>
> **Reviewer Comment 4:**
> "The graphlet approach uses no labels, but the homophily measurement does."
>
> **Our response:**
>
> We thank the reviewer for raising the model-selection question and we address it on three levels.
>
> First, we would like to clarify that homophily is the explanatory variable in our study — it accounts for why an embedding space is linearly separable — not a quantity we ask practitioners to compute to use the method. Our scientific claim (homophily is associated with separability) is distinct from the practical question of choosing a graphlet-based representation.
>
> Second, we agree with the reviewer that the selection of the corresponding graphlet representation is challenging. Please also note that embedding methods are all heuristics that are bound to fail on some instances (i.e., there is no “best for all”). For that reason, in practice, we rely on domain expertise to select the best graphlet representation. For instance, in social networks it is known that the triadic closure is the most frequent graph substructure [Bianconi et al., 2014] and in molecular networks that interecting genes form interconnected modules corresponding to dense graphlets (G2 and G8) [Sharan et al., 2007]. In addition, the representation coverage we introduce (fraction of nodes in the largest connected component of a representation) is a purely structural, label-free filter that discards fragmented high-order graphlet representations before any labels are seen.
>
> Third, pooling all graphlets into one embedding does not bypass the selection problem. To test this hypothesis, we add as an additional baseline HONE (Higher-Order Network Embedding) [Rossi et al., 2018], which learns jointly from all motif adjacencies a single embedding, so no graphlet need to be chosen. Yet in all the downstream tasks HONE is outperformed by an individual graphlet-based representation. Hence, concatenating all motifs into one does not resolve the selection problem.
>
> In the revised version of the manuscript, we add as a limitation that we do not provide an automatic, fully label-free rule for selecting the single best graphlet representation, instead we rely on domain expertise.
>
> ## References
>
> - Bianconi, Ginestra, et al. "Triadic closure as a basic generating mechanism of communities in complex networks." arXiv preprint arXiv:1407.1664 (2014).
> - Sharan, Roded, Igor Ulitsky, and Ron Shamir. "Network‐based prediction of protein function." Molecular systems biology 3.1 (2007)
> - Rossi, R. A., Ahmed, N. K., Koh, E., Kim, S., Rao, A., & Abbasi-Yadkori, Y. (2018). HONE: Higher-Order Network Embeddings. arXiv preprint arXiv:1801.09303.

---

> ### Author Response · Authors · 2026-07-03
> **Rebuttal Concern 5**
>
> **Reviewer Comment 5:**
> "Scalability is not discussed."
>
> **Our response:**
>
> We thank the reviewer and now report runtimes and the complexity analysis.
>
> 1. In our study we compute the graphlet adjacencies using the exact counter GRADCO [Windels et al., 2026], which has worst-case time complexity $O(n\,d^3)$, where $n$ is the number of nodes and $d$ is the maximum node degree. The cost does not blow up for a structural reason: computing up to four-node graphlets is sufficient because real networks are small-world (effective diameter 3–10 for 36 of the 40 networks in the benchmark of the GRADCO paper) and on networks of around 20,000 nodes GRADCO computes all orbit-adjacency matrices in minutes on a single CPU core. All computations are performed single-threaded and serially on an Intel Xeon E5-2124 CPU at 3.30 GHz with 128 GB of RAM.
>
> Every network in our study lies within this range: for example the largest multi-label network, human PPI (≈19k nodes), takes ≈8 min and human co-expression (≈17k) ≈9 min, while the single-label nets take seconds to a few minutes. Runtimes for all networks are given below.
>
> *Table — exact graphlet-counting runtime per network (single-threaded, Intel Xeon E-2124 @ 3.30 GHz, 128 GB RAM). Networks from Table 1 (single-label), the biological multi-label set, and the two larger heterophilic benchmarks added in the revision.*
>
> | Network | Domain | #Nodes | Exact GRADCO runtime |
> |---|---|---|---|
> | Amazon-ratings *(added in revision)* | heterophilic | 24,492 | 9.6 min |
> | Roman-empire *(added in revision)* | heterophilic | 22,662 | 7.7 min |
> | Human PPI | biological (multi-label) | ≈19,000 | ≈8 min |
> | CS Co-author | co-authorship | 18,333 | a few min |
> | Human co-expression | biological (multi-label) | ≈17,000 | ≈9 min |
> | Wikipedia CS | wiki | 10,657 | < 2 min |
> | Cora | citation | 2,708 | seconds |
> | Squirrel (filtered) | heterophilic | 2,223 | seconds |
> | CiteSeer | citation | 2,120 | seconds |
> | USA air-traffic | transport | 1,186 | seconds |
> | Chameleon (filtered) | heterophilic | 890 | seconds |
> | Yeast / pombe PPI and co-expression (4 nets) | biological (multi-label) | < human nets | < ≈9 min |
>
> After counting, the remaining stages are quadratic, not cubic: forming each $n\times n$ representation is $O(n^2)$ and the ONMTF factorisation is $O(n^2 k)$ per iteration with embedding dimension $k=\lceil\sqrt{n/2}\,\rceil\ll n$, after which the linear classifier is negligible.
>
> 2. To demonstrate that exact counting remains practical for larger networks above 20k nodes: we compute the graphlet-based representations exactly for two larger heterophilic benchmarks, Amazon-ratings (24,492 nodes) and Roman-empire (22,662), both in under 10 min. For larger networks, beyond ≈25k nodes, exact counting can be replaced by graphlet/orbit sampling, giving unbiased estimates at a cost set by the sample budget rather than exhaustive enumeration [Ribeiro et al., 2021]. Note that sampling is the same mechanism that makes random-walk embeddings scalable.
>
> ## References
>
> - Windels, S. F. L., Malod-Dognin, N., & Pržulj, N. (2026). Combining graphlets and random walks for capturing complex network topology. *Scientific Reports*.
> - Ribeiro, P., Paredes, P., Silva, M. E. P., Aparício, D., & Silva, F. (2021). A survey on subgraph counting: concepts, algorithms, and applications to network motifs and graphlets. *ACM Computing Surveys*, 54(2), Article 28, 1–36.

---

### Review · Reviewer_6KPH · 2026-05-12

**Summary Of Contributions:**

This paper introduces a graphlet-based network embedding framework that aims to simplify downstream analysis of complex networks by producing more linearly separable embedding spaces. The method extends standard random walk based embedding approaches, specifically DeepWalk and LINE, by replacing the standard adjacency matrix with graphlet adjacency matrices that capture both neighborhood and topological similarity of nodes simultaneously. The key insight is that more homophilic network representations yield more linearly separable embedding spaces, reducing the need for complex nonlinear models in downstream tasks. The approach is evaluated on thirteen networks spanning biological, social, citation and transportation domains, where it is compared against standard DeepWalk and LINE baseline.

**Audience:**

Yes

**Audience Explanation:**

Yes, the findings are relevant to the TMLR audience as understanding when and why network embedding spaces are linearly separable is a fundamental question of broad interest to the graph representation learning community.

**Claims And Evidence:**

No

**Claims Explanation:**

The evidence supporting the improvement of graphlet-based representations over the standard G0 baseline is not clearly presented. Table 1 only reports the maximum F1 score across all graphlets per method family, so it is impossible to directly see the gain over the G0 baseline. For biological networks, where the authors claim higher-order graphlets outperform G0, no G0 baseline scores are reported anywhere in the paper or the appendix. Supplementary Table 7 only identifies which graphlet achieved the maximum score but does not report the actual scores. The authors should report G0 scores explicitly for all datasets alongside the best graphlet scores. Furthermore, in cases where a higher-order graphlet outperforms G0, the authors should report whether this difference is statistically significant. Without these comparisons it is hard to conclude whether the proposed method provide any meaningful improvement over the standard baseline.

**Requested Changes:**

* \[1\] showed that there is significant label leakage in the Chameleon and Squirrel datasets. From the performance numbers it appears the authors used the non-filtered versions. Could the authors verify this and report results on the corrected filtered versions? Additionally, since these are the only two heterophilic datasets in the evaluation, it would be good to see performance on more heterophilic benchmarks. \[1\] has proposed larger and more reliable heterophilic datasets and results on those would strengthen the evaluation.
* Table 1 does not report any std. Could the authors run all experiments over multiple random seeds and report std or sem across seeds?
* All figures report std across datasets rather than across random seeds. This captures std in performance across datasets rather than the std of the method itself. Could the authors repeat all experiments across multiple random seeds and report std or sem across seeds instead?
* Could the authors consider using Laplacian eigenvectors as an additional baseline?
* The paper does not discuss the computational cost of computing graphlet adjacency matrices. Is the proposed approach feasible on larger real world graphs such as those in the Open Graph Benchmark (OGB)?

[1] Platonov, O., Kuznedelev, D., Diskin, M., Babenko, A., & Prokhorenkova, L. (2023). A critical look at the evaluation of GNNs under heterophily: Are we really making progress?. arXiv preprint arXiv:2302.11640.

---

> ### Author Response · Authors · 2026-07-03
> **Response to Requested Change 1 Part 1**
>
> **Reviewer Comment 1:**
> "Table 1 only reports the maximum F1 score across all graphlets per method family, so it is impossible to directly see the gain over the G0 baseline."
>
> **Our response:**
>
> We thank the reviewer for catching this. In the revised Table 1 we now report, for each method family, the best graphlet representation $k$ in parentheses (over the splits) and further comment on how often the higher-order graphlets ($k>0$) beat the standard adjacency-based baselines ($G_0$). We also add the requested Laplacian-eigenvector baseline — but rather than applying it only to the standard adjacency, we apply it over all nine graphlet adjacencies, to test whether higher-order structure improves this baseline too.
>
> *Table 1 — Per-family best node-classification $F_1$ over $k$ (best $k$ in parentheses) for L-SVM and RF, across all 15 networks; best per row in bold. Rows 1–9 are the single-label networks (Chameleon/Squirrel are the leakage-free filtered versions, Roman-empire/Amazon-ratings the larger Platonov heterophilic benchmarks; WikiCS uses the 60/20/20 ten-split), rows 10–15 the biological multi-label networks. $G_0$ is the standard (non-graphlet) representation.*
>
> | Networks | DeepWalk L-SVM | DeepWalk RF | LINE L-SVM | LINE RF | A L-SVM | A RF | Laplacian L-SVM | Laplacian RF |
> |---|---|---|---|---|---|---|---|---|
> | Cora | 0.827±0.017 (k=0) | **0.840±0.011 (k=0)** | 0.792±0.010 (k=3) | 0.822±0.017 (k=1) | 0.754±0.010 (k=3) | 0.789±0.017 (k=1) | 0.832±0.015 (k=0) | 0.830±0.016 (k=0) |
> | Wikipedia CS | **0.822±0.006 (k=0)** | 0.816±0.006 (k=0) | 0.778±0.007 (k=3) | 0.777±0.009 (k=0) | 0.740±0.004 (k=0) | 0.761±0.008 (k=0) | 0.813±0.006 (k=0) | 0.811±0.007 (k=0) |
> | CS Co-author | **0.901±0.002 (k=0)** | 0.895±0.003 (k=0) | 0.853±0.004 (k=3) | 0.851±0.003 (k=3) | 0.834±0.005 (k=3) | 0.835±0.003 (k=1) | 0.881±0.004 (k=0) | 0.882±0.003 (k=0) |
> | USA air-traffic | 0.636±0.023 (k=1) | 0.648±0.025 (k=3) | 0.674±0.019 (k=7) | **0.684±0.024 (k=2)** | 0.663±0.026 (k=2) | 0.680±0.025 (k=0) | 0.664±0.026 (k=0) | 0.667±0.028 (k=0) |
> | Chameleon (filt.) | 0.388±0.047 (k=6) | 0.406±0.027 (k=0) | 0.393±0.026 (k=0) | 0.405±0.033 (k=0) | 0.367±0.029 (k=1) | 0.386±0.035 (k=0) | 0.393±0.039 (k=0) | **0.440±0.028 (k=0)** |
> | CiteSeer | 0.704±0.027 (k=1) | **0.721±0.018 (k=0)** | 0.672±0.022 (k=1) | 0.709±0.016 (k=1) | 0.663±0.020 (k=3) | 0.695±0.017 (k=3) | 0.707±0.024 (k=0) | 0.709±0.021 (k=0) |
> | Squirrel (filt.) | 0.359±0.019 (k=3) | 0.358±0.029 (k=3) | 0.375±0.026 (k=0) | **0.391±0.017 (k=0)** | 0.337±0.013 (k=3) | 0.361±0.022 (k=0) | 0.381±0.023 (k=3) | 0.373±0.018 (k=0) |
> | Roman-empire | 0.133±0.005 (k=2) | 0.131±0.004 (k=2) | 0.146±0.002 (k=0) | 0.177±0.003 (k=0) | 0.138±0.004 (k=2) | **0.179±0.004 (k=0)** | 0.130±0.004 (k=1) | 0.120±0.004 (k=1) |
> | Amazon-ratings | 0.512±0.006 (k=6) | **0.517±0.006 (k=6)** | 0.503±0.005 (k=1) | 0.512±0.005 (k=3) | 0.492±0.005 (k=3) | 0.509±0.005 (k=3) | 0.449±0.007 (k=3) | 0.507±0.006 (k=3) |
> | Pombe COEX | 0.527±0.003 (k=8) | 0.507±0.003 (k=0) | 0.527±0.003 (k=7) | 0.500±0.003 (k=3) | 0.530±0.002 (k=2) | 0.497±0.004 (k=7) | **0.533±0.001 (k=8)** | 0.513±0.001 (k=2) |
> | Cerevisiae COEX | 0.606±0.001 (k=2) | 0.590±0.002 (k=0) | 0.610±0.001 (k=3) | 0.582±0.002 (k=3) | 0.602±0.001 (k=6) | 0.579±0.001 (k=4) | **0.613±0.001 (k=0)** | 0.593±0.003 (k=0) |
> | Homo sapiens COEX | 0.515±0.001 (k=8) | 0.499±0.002 (k=2) | 0.520±0.001 (k=1) | 0.498±0.001 (k=6) | 0.517±0.001 (k=2) | 0.497±0.001 (k=7) | **0.529±0.001 (k=2)** | 0.510±0.001 (k=8) |
> | Pombe PPI | 0.661±0.002 (k=0) | 0.626±0.003 (k=0) | 0.588±0.002 (k=6) | 0.592±0.004 (k=2) | 0.563±0.005 (k=2) | 0.581±0.005 (k=2) | **0.663±0.003 (k=0)** | 0.645±0.004 (k=0) |
> | Cerevisiae PPI | 0.647±0.002 (k=2) | 0.626±0.002 (k=8) | 0.635±0.002 (k=0) | 0.615±0.001 (k=8) | 0.606±0.001 (k=2) | 0.601±0.001 (k=8) | **0.677±0.002 (k=0)** | 0.649±0.001 (k=0) |
> | Homo sapiens PPI | 0.528±0.001 (k=2) | 0.490±0.002 (k=8) | 0.524±0.001 (k=8) | 0.485±0.001 (k=8) | 0.509±0.002 (k=8) | 0.477±0.001 (k=8) | **0.536±0.001 (k=2)** | 0.512±0.001 (k=2) |

---

> > ### Author Response · Authors · 2026-07-03
> > **Response to Requested Change 1 Part 2**
> >
> > **How often do higher-order graphlets ($k>0$) beat the standard adjacency ($G_0$)?** We report this in two ways: (i) *global winner*, i.e. the single best cell per network, over both classifiers and all four methods; and (ii) *per embedding family*, i.e. in how many of the 15 networks that family's best (over L-SVM and RF) is at $k>0$.
> >
> > Under (i), a higher graphlet order is best in **5 of 15 networks** (USA air-traffic, Amazon-ratings, and three biological nets). This global count understates the effect, since it is decided by whichever single method happens to peak on a given network. Under (ii), higher-order graphlets win the majority of networks for every graphlet-extended family as shown in the following Table.
> >
> > *Per-family count: number of the 15 networks on which each family's best result (over L-SVM and RF) is achieved at $G_0$ ($k=0$) vs at a higher order ($k>0$); last column: on those higher-order wins, the average F1 improvement of the best $k>0$ over the same family's $G_0$.*
> >
> > | Embedding family | best at $G_0$ ($k=0$) | best at higher order ($k>0$) | avg. F1 gain over $G_0$ (on the $k>0$ wins) |
> > |---|---|---|---|
> > | DeepWalk$_{G_k}$ | 6 | **9** | +0.017 |
> > | LINE$_{G_k}$ | 4 | **11** | +0.013 |
> > | $A_{G_k}$ (graphlet adjacency) | 5 | **10** | +0.016 |
> > | Laplacian (over $A_{G_k}$) | 9 | 6 | +0.010 |
> >
> > To conclude, while $G_0$ remains best on the homophilic networks where the standard adjacency is already the most informative representation — the higher-order graphlet adjacencies improve the result on the majority of networks for each of our graphlet-extended embeddings (LINE${G_k}$ 11/15, $A{G_k}$ 10/15, DeepWalk${G_k}$ 9/15), and even improve the standard Laplacian baseline in 6/15. In addition, on the networks where a higher order graphlet adjacency wins, the gain over that family's own $G_0$ is consistent, averaging $+0.017$ (DeepWalk${G_k}$), $+0.013$ (LINE${G_k}$), $+0.016$ ($A{G_k}$) and $+0.010$ (Laplacian) in $F_1$-score, and reaching up to $+0.055$ on individual networks. The graphlet-based extensions therefore add value precisely on the networks whose structure is not fully captured by the standard adjacency.

---

> ### Author Response · Authors · 2026-07-03
> **Response to Comment 2 Part 1**
>
> **Reviewer Comment 2:**
> "[1] showed that there is significant label leakage in the Chameleon and Squirrel
> datasets. From the performance numbers it appears the authors used the non-filtered versions. Could the authors verify this and report results on the corrected filtered versions? Additionally, since these are the only two heterophilic datasets in the evaluation, it would be good to see performance on more heterophilic benchmarks. [1] has proposed larger and more reliable heterophilic datasets and results on those would strengthen the evaluation."
>
> **Our response:**
>
> We thank the reviewer and confirm that the original evaluation used the non-filtered Chameleon and Squirrel datasets. We have corrected this throughout: all heterophilic results now use the leakage-free filtered versions of Platonov et al. (2023) [1]. We further extend the evaluation to the two largest Platonov heterophilic benchmarks, Amazon-ratings (24,492 nodes) and Roman-empire (22,662 nodes); the revised statistics table now includes all four. Following Reviewer bu8G and g9af, we also add GCN, Motif-GNN, GAT and HONE (Higher-Order Network Embedding) [2], which learns a single embedding jointly from all motif adjacencies, as baselines.
>
> *Filtered and larger heterophilic benchmarks — best $F_1$ over graphlet representation $k$ and classifier (L-SVM/RBF/RF), best graphlet-representation $k$ in parentheses for the graphlet-embeddings; best per network in bold. The GNN embeddings use the datasets' node features and the Laplacian Eigenmaps and HONE embeddings are structure-only.*
>
> | Network ($N$) | DeepWalk$_{G_k}$ | LINE$_{G_k}$ | $A_{G_k}$ | Laplacian | HONE | GCN | GAT | Motif-GNN |
> |---|---|---|---|---|---|---|---|---|
> | Chameleon-filt. (890) | 0.406 (k=0) | 0.405 (k=0) | 0.386 (k=0) | **0.440 (k=0)** | 0.416 | 0.433 | 0.404 | 0.412 |
> | Squirrel-filt. (2223) | 0.359 (k=3) | 0.391 (k=0) | 0.361 (k=0) | 0.381 (k=3) | 0.379 | 0.379 | 0.332 | **0.432** |
> | Roman-empire (22662) | 0.133 (k=2) | 0.177 (k=0) | 0.179 (k=0) | 0.134 (k=1) | 0.113 | 0.527 | 0.592 | **0.606** |
> | Amazon-ratings (24492) | **0.517 (k=6)** | 0.512 (k=3) | 0.509 (k=3) | 0.507 (k=3) | 0.510 | 0.386 | 0.399 | 0.230 |
>
> On the small filtered datasets (Chameleon, Squirrel), no method dominates: structure-only embeddings and feature-using GNNs are comparable, and the best result is split between the two families — the purely structural Laplacian eigenvectors win on Chameleon, while the Motif-GNN wins on Squirrel. The two larger benchmarks are more informative, because they dissociate structural signal from feature signal. Roman-empire is a very sparse network (density = 0.00013) with essentially no structural signal (node homophily ≈ 0.046), so a structure-only representation is expected to be near-random. This is exactly what we observe (F1-scores in the range of 0.13 to 0.18), whereas the feature-using models recover the labels from node features and yield F1-scores in the range of 0.53 to 0.61. On Amazon-ratings, where the structural signal is present, the structure-only embeddings (0.52) instead outperform all three GNNs (0.23–0.40).

---

> > ### Author Response · Authors · 2026-07-03
> > **Response to Comment 2 Part 2**
> >
> > Following Reviewer bu8G's suggestion, we have also applied the additional baselines in the biological networks that do not have node features; so both our embeddings and the GNNs use only structural information here. For completenes / ease of reading, we reproduce our response here.
> >
> > **2. Feature-free biological networks: structure wins.** The GNNs are given feature-free structural inputs: a learned node-identity input (results are comparable with $\log_2$-binned degree features). The GNNs are trained end-to-end with a per-class binary-cross-entropy objective and evaluated by weighted $F_1$ over 10 seeds. When features are not available, there is a graphlet-based embedding coupled with a simple linear classifier that outperform all three GNNs on every network. Notably, the structure-only HONE baseline also outperforms every GNN here while staying below our graphlet embeddings throughout. This shows that the advantage of structure-based embeddings on feature-free data is robust, and that our graphlet embeddings are the strongest among them.
> >
> > *Table — biological (multi-label) networks, weighted $F_1$ (mean ± std over 10 seeds). Best non-NN embedding = best over the three ONMTF families (best classifier in $\{$L-SVM, RBF-SVM, RF$\}$; winning family in parentheses); bold = best per row.*
> >
> > | Network | Best non-NN embedding | HONE | GCN | GAT | Motif-GNN |
> > |---|---|---|---|---|---|
> > | PPI (human) | **0.528 (DeepWalk$_{G_2}$)** | 0.513 | 0.449 | 0.452 | 0.456 |
> > | PPI (yeast) | **0.647 (DeepWalk$_{G_2}$)** | 0.644 | 0.529 | 0.514 | 0.472 |
> > | PPI (pombe) | **0.661 (DeepWalk$_{G_0}$)** | 0.611 | 0.525 | 0.552 | 0.434 |
> > | Co-exp (human) | **0.524 (DeepWalk$_{G_8}$)** | 0.505 | 0.463 | 0.455 | 0.447 |
> > | Co-exp (yeast) | **0.610 (LINE$_{G_3}$)** | 0.598 | 0.512 | 0.514 | 0.489 |
> > | Co-exp (pombe) | **0.530 (A$_{G_2}$)** | 0.520 | 0.454 | 0.468 | 0.452 |
> >
> > These comparisons place our work more precisely in the context of heterophilic networks and GNNs, and we will revise the manuscript accordingly: when informative node features are available, GNNs perform better than the feature-free methods. On the other hand, when features are not available or uninformative, as in the case of Amazon-ratings, and the signal is instead encoded in the network structure, purely structural graphlet-based embeddings with a simple linear classifier match or outperform end-to-end GNNs. This is common in molecular and many other real-world networks (e.g, transport networks), where such structure-based embeddings alleviate the need for GNNs.
> >
> > ## References
> >
> > [1] Platonov, O., Kuznedelev, D., Diskin, M., Babenko, A., & Prokhorenkova, L. (2023). A critical look at the evaluation of GNNs under heterophily: Are we really making progress? *arXiv preprint* arXiv:2302.11640.
> > [2] Rossi, R. A., Ahmed, N. K., Koh, E., Kim, S., Rao, A., & Abbasi-Yadkori, Y. (2018). HONE: Higher-Order Network Embeddings. *arXiv preprint* arXiv:1801.09303.

---

> ### Author Response · Authors · 2026-07-03
> **Response to Requested Change 3**
>
> **Reviewer Comment 3:**
> "Table 1 does not report any std. Could the authors run all experiments over multiple random seeds and report std or sem across seeds?" / "All figures report std across datasets rather than across random seeds. … Could the authors repeat all experiments across multiple random seeds and report std or sem across seeds instead?"
>
> **Our response:**
>
> We thank the reviewer for the suggestion and have revised all reported variability to reflect split/seed variability of the methods rather than variability across datasets.
>
> For the single-label networks, we report mean ± std over the canonical evaluation splits where they exist — the GeomGCN 10 splits (Cora, CiteSeer), the 20 WikiCS splits, and the 10 leakage-free Platonov splits (filtered Chameleon, Squirrel) — which we adopt for comparability with prior work; for the two networks without a canonical split (CS Co-author, USA air-traffic) we use 10 fixed-seed random 50/25/25 splits.
>
> For the biological networks, we repeat the 10-fold cross-validation over 10 random seeds and report mean ± std across seeds.
>
> In the revised manuscript, all bar plots show error bars over these splits/seeds (i.e. the variability of the method itself), replacing the previous std across datasets.
>
> ---

---

> ### Author Response · Authors · 2026-07-03
> **Response to Requested Change 4**
>
> **Reviewer Comment 4:**
> "Could the authors consider using Laplacian eigenvectors as an additional baseline?"
>
> **Our response:**
>
> We agree with the reviewer that adding another structural baseline, such as Laplacian eigenvectors, helps to show the benefits of our method. Following Reviewer g9af, we also add a further structural baseline, HONE (Higher-Order Network Embedding) [2], which learns a single embedding jointly from all motif adjacencies. To make the comparison informative, we apply the Laplacian-eigenvector baseline not only to the standard adjacency but over all nine graphlet adjacencies $A_{G_k}$ ($k=0..8$, where $k{=}0$ is the standard adjacency), to test whether higher-order structure improves this baseline too. The Laplacian Eigenmaps $F_1$ results (L-SVM / RBF / RF) are reported in Table 1, and the information-retrieval results (weighted AUROC) in the following table.
>
> *Cosine-weighted AUROC for the embedding methods (downstream retrieval task); best over $k$ with the corresponding $k$ in parentheses (HONE reports the best variant); best per network in bold.*
>
> | Network | DeepWalk$_{G_k}$ | LINE$_{G_k}$ | $A_{G_k}$ | Laplacian | HONE |
> |---|---|---|---|---|---|
> | Cora | **0.782 (k=3)** | 0.722 (k=3) | 0.726 (k=3) | 0.744 (k=1) | 0.740 |
> | CiteSeer | 0.650 (k=3) | 0.611 (k=3) | **0.654 (k=3)** | 0.653 (k=0) | 0.618 |
> | Wikipedia CS | **0.801 (k=0)** | 0.765 (k=3) | 0.715 (k=1) | 0.729 (k=0) | 0.773 |
> | CS Co-author | **0.880 (k=0)** | 0.841 (k=3) | 0.827 (k=1) | 0.831 (k=0) | 0.846 |
> | USA air-traffic | 0.594 (k=8) | 0.567 (k=8) | **0.596 (k=7)** | 0.545 (k=0) | 0.549 |
> | Chameleon-filt. | **0.544 (k=4)** | 0.533 (k=0) | 0.540 (k=8) | 0.541 (k=6) | 0.538 |
> | Squirrel-filt. | **0.521 (k=3)** | 0.513 (k=0) | 0.510 (k=4) | 0.519 (k=3) | 0.509 |
> | Amazon-ratings | 0.512 (k=0) | 0.512 (k=4) | 0.516 (k=5) | **0.521 (k=0)** | 0.512 |
> | **Average** | **0.660** | 0.633 | 0.636 | 0.635 | 0.636 |
>
> On the information-retrieval task, our graphlet-based extension of the DeepWalk embeddings (DeepWalk${G_k}$) achieves the highest average AUROC (0.660) and is the best method on 5 of 8 networks, ahead of both the Laplacian baseline (0.635) and HONE (0.636); on two of the remaining three, the top result is still a graphlet-based embedding ($A{G_k}$, on CiteSeer and USA air-traffic), and only on Amazon-ratings — where all methods are on par (AUROC $\approx$ 0.51–0.52) — does the Laplacian baseline edge marginally ahead. We further observe that the Laplacian baseline itself improves under our graphlet-based extensions: its best result is at a higher order ($k>0$) on several networks (e.g. Cora $k=1$, Chameleon $k=6$, Squirrel $k=3$), so higher-order structural information benefits even a standard spectral method. Finally, HONE, which learns a single embedding jointly from all motif adjacencies, does not outperform our individual graphlet-based representations. This indicates that fusing all higher-order information into one embedding does not, on its own, improve over selecting an appropriate graphlet representation.
>
> Finally, we also compare the structural baselines with our methods on the **functional module discovery** task on the six biological networks. Recall that in this downstream task we first cluster the genes in the embedding space (KMeans, $\sqrt{N/2}$ clusters) and test each cluster (module) for enrichment of Reactome pathways; *functional coverage* is the fraction of annotated Reactome pathways that are significantly enriched in at least one module, i.e. how much of the known biology the embedding organises into coherent modules.
>
> *Reactome functional coverage (%) from the discovered modules; best over $k$ with the corresponding $k$ in parentheses (HONE reports the best variant); best per network in bold.*
>
> | Network | DeepWalk$_{G_k}$ | LINE$_{G_k}$ | $A_{G_k}$ | Laplacian | HONE |
> |---|---|---|---|---|---|
> | Pombe COEX | **48.5 (k=0)** | 43.6 (k=1) | 41.5 (k=4) | 45.6 (k=7) | 39.5 |
> | Cerevisiae COEX | **63.3 (k=7)** | 62.1 (k=1) | 58.3 (k=5) | 59.1 (k=1) | 47.1 |
> | Homo sapiens COEX | 61.5 (k=2) | 65.6 (k=5) | 63.9 (k=8) | **66.1 (k=7)** | 54.5 |
> | Pombe PPI | **50.6 (k=0)** | 33.1 (k=3) | 25.6 (k=8) | 36.5 (k=5) | 34.5 |
> | Cerevisiae PPI | **57.3 (k=2)** | 45.6 (k=0) | 34.0 (k=2) | 52.2 (k=0) | 49.6 |
> | Homo sapiens PPI | **73.6 (k=8)** | 69.7 (k=8) | 63.8 (k=8) | 52.3 (k=2) | 62.7 |
> | **Average** | **59.1** | 53.3 | 47.9 | 52.0 | 48.0 |
>
> On this downstream task, our graphlet-based extensions of DeepWalk (DeepWalk$_{G_k}$) capture the most biological signal: it has the highest functional coverage on five of the six networks and the highest average (59.1\%), clearly outperforming both the Laplacian (52.0\%) and HONE (48.0\%) baselines.
>
> ## References
>
> [2] Rossi, R. A., Ahmed, N. K., Koh, E., Kim, S., Rao, A., & Abbasi-Yadkori, Y. (2018). HONE: Higher-Order Network Embeddings. *arXiv preprint* arXiv:1801.09303

---

> ### Author Response · Authors · 2026-07-03
> **Response to Requested Change 5:**
>
> **Reviewer Comment 5:**
> "The paper does not discuss the computational cost of computing graphlet adjacency matrices. Is the proposed approach feasible on larger real world graphs such as those in the Open Graph Benchmark (OGB)?."
>
> **Our response:**
>
> We thank the reviewer for the comment and now report runtimes and a complexity analysis. Please note that the same point was raised by Reviewer bu8G; for ease of reading, we reproduce our response here.
>
> 1. In our study we compute the graphlet adjacencies using the exact counter GRADCO [3], which has worst-case time complexity $O(n\,d^3)$, where $n$ is the number of nodes and $d$ is the maximum node degree. The cost does not blow up for a structural reason: computing up to four-node graphlets is sufficient because real networks are small-world (effective diameter 3–10 for 36 of the 40 networks in the benchmark of the GRADCO paper) and on networks of around 20,000 nodes GRADCO computes all orbit-adjacency matrices in minutes on a single CPU core.
>
> Every network in our study lies within this range: for example the largest multi-label network, human PPI (≈19k nodes), takes ≈8 min and human co-expression (≈17k) ≈9 min, while the single-label nets take seconds to a few minutes.
>
> *Table — exact graphlet-counting runtime per network (single-threaded, Intel Xeon E-2124 @ 3.30 GHz, 128 GB RAM). Networks from Table 1 (single-label), the biological multi-label set, and the two larger heterophilic benchmarks added in the revision.*
>
> | Network | Domain | #Nodes | Exact GRADCO runtime |
> |---|---|---|---|
> | Amazon-ratings *(added in revision)* | heterophilic | 24,492 | 9.6 min |
> | Roman-empire *(added in revision)* | heterophilic | 22,662 | 7.7 min |
> | Human PPI | biological (multi-label) | ≈19,000 | ≈8 min |
> | CS Co-author | co-authorship | 18,333 | a few min |
> | Human co-expression | biological (multi-label) | ≈17,000 | ≈9 min |
> | Wikipedia CS | wiki | 10,657 | < 2 min |
> | Cora | citation | 2,708 | seconds |
> | Squirrel (filtered) | heterophilic | 2,223 | seconds |
> | CiteSeer | citation | 2,120 | seconds |
> | USA air-traffic | transport | 1,186 | seconds |
> | Chameleon (filtered) | heterophilic | 890 | seconds |
> | Yeast / pombe PPI and co-expression (4 nets) | biological (multi-label) | < human nets | < ≈9 min |
>
> 2. To demonstrate that exact counting remains practical for larger networks above 20k nodes: we compute the graphlet-based representations exactly for two larger heterophilic benchmarks, Amazon-ratings (24,492 nodes) and Roman-empire (22,662), both in under 10 min. For larger networks, like the ones in the Open Graph Benchmark (OGB) that consist of million of nodes, exact counting can be replaced by graphlet/orbit sampling, giving unbiased estimates at a cost set by the sample budget rather than exhaustive enumeration [4]. Note that sampling is the same mechanism that makes random-walk embeddings scalable.
>
> ---
>
> ## References
> [3] Windels, S. F. L., Malod-Dognin, N., & Pržulj, N. (2026). GRADCO: a graphlet-orbit adjacency counter for efficient analysis of large networks. *Scientific Reports*, 16, 14902.
>
> [4] Ribeiro, P., Paredes, P., Silva, M. E. P., Aparício, D., & Silva, F. (2021). A survey on subgraph counting. *ACM Computing Surveys*, 54(2), Article 28.

---

### Review · Reviewer_g9af · 2026-06-22

**Summary Of Contributions:**

# Summary

* This paper studies when network embedding spaces become linearly separable and proposes graphlet-based network matrix representations to improve the homophily of the input representation. Specifically, the authors construct graphlet adjacency matrices and use them to extend the closed-form matrix formulations of LINE and DeepWalk. The resulting matrices are then factorized to obtain node embeddings, which are evaluated using linear SVM, RBF SVM, Random Forest, cosine-similarity-based retrieval, and functional module discovery tasks. The central claim is that more homophilic network matrix representations tend to yield more linearly separable embedding spaces and better downstream performance.

# Strengths
* The paper addresses an important issue in graph/network representation learning.
* The idea of studying the relationship between homophily of input matrix representations and linear separability of the resulting embedding spaces is interesting.
* The proposed graphlet-based extensions of LINE and DeepWalk are easy to understand and provide an interpretable way to incorporate higher-order local structures into network embeddings.

# Weaknesses
* The title is too broad. The title seems to include all machine learning fields while the core discussion focuses on graph representation learning.
* The motivation is not convincing. The paper motivates the study using linear semantic relationships in NLP/VLM representations, but the actual analysis focuses on linear separability of node labels via linear classifiers. These are related but distinct notions.
* The technical novelty appears limited. The core technical operation is to replace the original adjacency matrix in existing LINE/DeepWalk closed-form PPMI formulations with graphlet adjacency matrices, followed by standard matrix factorization. This is a natural and potentially useful combination, but it seems incremental.
* The technical route is closely related to prior work on higher-order network embeddings[1], motif-based adjacency construction[2], graphlet Laplacians, and NetMF-style matrix factorization。 The paper should more clearly distinguish its contribution from these lines of work and include stronger baselines from higher-order/motif/graphlet embedding methods.
  * [1] HONE: Higher-Order Network Embeddings
  * [2] Motif Graph Neural Network
* There is a lack of theoretical analysis. The paper now only shows some empirical results on the linear separability.  It could be useful if the author can show "why the homophilic graphlet-based representation can improve the linear separability" theoretically.
* The proposed graphlet-based representation appears to rely on a strong structural assumption: local higher-order subgraph patterns should correlate with node labels or semantic similarity. It is not clear whether it holds for graphs where labels are mainly determined by node attributes, long-range/global structures, edge types, temporal dynamics, or heterogeneous relations.
* If a network structure is with a strong assumption on motif/graphlet semantics, the problem appears to be easy. Maybe existing technology such as HONE can fully sovle the problem.
* The scalability of the proposed framework should be analyzed. Networks in the real world could be very large, such as social network. The computational complexity for graphlet construcation could too expensive to run the algorithm. It may further makes the whole process more complex and finally contradicts the title "simplifying complex machinle learning ..."
* The proposed framework generates multiple graphlet-based matrix representations and often reports the best-performing representation for each dataset or task. There is no a general rule for readers to select the best representation. Select the best-performing graphlet representation using label information may introduce model-selection bias. It could overfit on the training set.

**Additional Comments:**

Based on my comments above, I suggest a weak rejection.

**Audience:**

Yes

**Audience Explanation:**

Researchers working on graph representation learning, network embedding, heterophily, and biological network analysis would be interested in this paper’s findings.  The paper studies an interesting question: whether more homophilic graph representations lead to more linearly separable embedding spaces. It provides empirical evidence across multiple real and synthetic networks.

**Broader Impact Concerns:**

A brief broader impact statement could still mention that network embedding methods may be applied to biological or social networks, where privacy, fairness, and over-interpretation of predicted relationships should be considered. However, I do not see any substantial ethical concern that would materially affect my evaluation of the paper.

**Claims And Evidence:**

No

**Claims Explanation:**

The submission makes several major claims, and while some of them are supported by empirical evidence, the overall support is not fully convincing for the broader claims made in the paper.

- The construction of the proposed representation is clearly described. The authors replace the original adjacency matrix in LINE/DeepWalk-style formulations with graphlet adjacency matrices and then factorize the resulting matrices to obtain node embeddings. This supports the claim that the method incorporates higher-order graph structures.
- The claim that graphlet-based representations can be more homophilic is supported by experiments on 6 biological multi-label networks and 7 single-label networks. The reported node homophily, edge homophily, and GSI results show that at least one graphlet-based representation improves over the standard adjacency-based baselines. However, this should be viewed as an empirical observation on the tested datasets rather than a general guarantee.
- The authors claim that more homophilic input matrix representations lead to more linearly separable embedding spaces. The paper provides evidence through node classification experiments with linear SVM, RBF SVM, and Random Forest. This is an interesting empirical finding. However, the evidence is mainly correlational. Linear separability is measured through linear SVM performance, which is a useful proxy but does not fully characterize the geometry of the embedding space. The paper does not provide deeper analysis of margins, intra-/inter-class distances, spectral properties, etc. Therefore, the evidence supports an empirical association, but not a strong mechanistic or theoretical claim.
- The paper claims that such linearly separable embedding spaces can reduce the need for complex machine learning models in downstream analysis. This claim is only partially supported. The comparison between linear and non-linear classifiers provides some evidence that linear methods can work well in several embedding spaces. However, the paper does not sufficiently analyze the computational cost of the full pipeline, including graphlet enumeration, construction of multiple graphlet adjacency matrices, matrix transformations, and matrix factorization. Thus, the claim about simplifying complex machine learning is not fully supported from a computational perspective.

**Requested Changes:**

I would request the following changes.

1. **Clarify the scope and framing. Critical.**
   Revise the title, abstract, and introduction to reflect that the paper is mainly about graph/network representation learning, not machine learning in general.

2. **Clarify the motivation around linearity. Critical.**
   Distinguish linear semantic relationships in NLP/VLMs from linear separability of node labels in graph embeddings.

3. **Provide theoretical analysis of linear separability. Critical.**
   Add theoretical or formal analysis explaining why more homophilic graphlet-based representations should improve the linear separability of the resulting embedding space.

4. **Better position the method against prior work. Critical.**
   Clearly explain how the method differs from higher-order network embeddings, motif-based methods, graphlet Laplacians, GDV-based embeddings, and NetMF-style methods.

5. **Add stronger baselines. Critical.**
   Include representative higher-order / motif / graphlet embedding baselines, such as HONE or motif-based graph methods.

6. **Address graphlet selection bias. Critical.**
   Provide a principled rule for selecting graphlet representations, or use a strict validation/nested cross-validation protocol. Report fixed, average, and best graphlet results separately.

7. **Analyze scalability. Critical.**
   Report runtime, memory cost, graph sizes, and complexity of graphlet construction, matrix transformation, and factorization.

8. **Clarify assumptions and limitations. Critical.**
   Discuss when local graphlet/motif structures are expected to correlate with labels, and when the method may fail.

9. **Deepen the empirical analysis of linear separability. Would strengthen the work.**
   Add analysis of margins, embedding geometry, inter-/intra-class distances, spectral properties, or robustness to different splits.

---

> ### Author Response · Authors · 2026-07-03
> **Requested Change 1**
>
> We thank the reviewer for recognising the importance of the problem and the
> interpretability of the approach, and for the constructive suggestions. We address each point below.
>
> **Reviewer Comment 1:**
> "Clarify the scope and framing. Critical. Revise the title, abstract, and introduction to reflect that the paper is mainly about graph/network representation learning, not machine learning in general."
>
> **Our response:**
>
> We agree that the framing was too broad, and we have retitled the paper to scope it precisely to graph representation learning:
>
>  **"Simplifying graph representation learning by linearly separable network embedding spaces."**
>
> We also agree that the motivation in our Introduction around linearity was under-explained (Reviewer Comment 2); we address this in detail in our response to that comment. We have revised the abstract and introduction accordingly to make clear that the paper concerns graph/network representation learning specifically, rather than machine learning in general.
>
> **Revised Abstract:**
>
> Low-dimensional embeddings are a common way to represent complex networks, but we still do not fully understand how they organise information. One idea, called the linear representation hypothesis, suggests that important concepts are encoded as linear directions or subspaces in the embedding space. In language models and vision–language models, this makes it possible to uncover meaningful relationships using simple linear operations on the embedding vectors. Linear separability offers a direct way to test this idea: if the nodes of a given class can be separated from the rest by a straight line (or, in higher dimensions, a flat boundary),  then that class is represented in a simple linear way. This raises an important question: do network embeddings share this structure, with node classes separable by simple linear boundaries, making them easy to identify and exploit? In this study, we gain fundamental insight into the structure of network data that gives rise to this linearity, showing that the more homophilic the network representation, the more linearly separable the corresponding network embedding space, which in turn yields better downstream analysis results. We demonstrate the applicability of our insight on thirteen networks from multiple domains: six multi-label biological networks and seven single-label networks from the social, citation, and transportation domains. These insights into the structure of network data that enables its linear mining and exploitation provide a foundation to build upon for efficient and explainable mining of complex network data.

---

> > ### Author Response · Authors · 2026-07-03
> > **Requested Change 7**
> >
> > **Reviewer Comment 7:**
> > "Analyze scalability. Critical. Report runtime, memory cost, graph sizes, and complexity of graphlet construction, matrix transformation, and factorization."
> >
> > **Our response:**
> >
> > We thank the reviewer for the comment and now report runtimes and a complexity analysis. Please note that the same point was raised by Reviewer bu8G and 6KPH; for ease of reading, we reproduce our response here.
> >
> > 1. In our study we compute the graphlet adjacencies using the exact counter GRADCO [12], which has worst-case time complexity $O(n\,d^3)$, where $n$ is the number of nodes and $d$ is the maximum node degree. The cost does not blow up for a structural reason: computing up to four-node graphlets is sufficient because real networks are small-world (effective diameter 3–10 for 36 of the 40 networks in the benchmark of the GRADCO paper) and on networks of around 20,000 nodes GRADCO computes all orbit-adjacency matrices in minutes on a single CPU core. All computations were performed single-threaded and serially on an Intel Xeon E5-2124 CPU at 3.30 GHz with 128 GB of RAM and were reproduced on a laptop with 24 GB of RAM.
> >
> > Every network in our study lies within this range: for example the largest multi-label network, human PPI (≈19k nodes), takes ≈8 min and human co-expression (≈17k) ≈9 min, while the single-label nets take seconds to a few minutes. Runtimes for all networks are given below.
> >
> > *Table — exact graphlet-counting runtime per network (single-threaded, Intel Xeon E-2124 @ 3.30 GHz, 128 GB RAM). Networks from Table 1 (single-label), the biological multi-label set, and the two larger heterophilic benchmarks added in revision.*
> >
> > | Network | Domain | #Nodes | Exact GRADCO runtime |
> > |---|---|---|---|
> > | Amazon-ratings *(added in revision)* | heterophilic | 24,492 | 9.6 min |
> > | Roman-empire *(added in revision)* | heterophilic | 22,662 | 7.7 min |
> > | Human PPI | biological (multi-label) | ≈19,000 | ≈8 min |
> > | CS Co-author | co-authorship | 18,333 | a few min |
> > | Human co-expression | biological (multi-label) | ≈17,000 | ≈9 min |
> > | Wikipedia CS | wiki | 10,657 | < 2 min |
> > | Cora | citation | 2,708 | seconds |
> > | Squirrel (filtered) | heterophilic | 2,223 | seconds |
> > | CiteSeer | citation | 2,120 | seconds |
> > | USA air-traffic | transport | 1,186 | seconds |
> > | Chameleon (filtered) | heterophilic | 890 | seconds |
> > | Yeast / pombe PPI and co-expression (4 nets) | biological (multi-label) | < human nets | < ≈9 min |
> >
> > After graphlet counting, the remaining stages are quadratic, not cubic: forming each $n\times n$ representation is $O(n^2)$ and the ONMTF factorisation is $O(n^2 k)$ per iteration with embedding dimension $k=\lceil\sqrt{n/2}\,\rceil\ll n$, after which the linear classifier is negligible.
> >
> > 2. To demonstrate that exact counting remains practical for larger networks above 20k nodes: we compute the graphlet-based representations exactly for two larger heterophilic benchmarks, Amazon-ratings (24,492 nodes) and Roman-empire (22,662), both in under 10 min. For larger networks, beyond ≈25k nodes, exact counting can be replaced by graphlet/orbit sampling, giving unbiased estimates at a cost set by the sample budget rather than exhaustive enumeration [13]. Note that sampling is the same mechanism that makes random-walk embeddings scalable.
> >
> > ### References
> >
> > [12] Windels, S. F. L., Malod-Dognin, N., & Pržulj, N. (2026). Combining graphlets and random walks for capturing complex network topology. *Scientific Reports*. DOI: 10.1038/s41598-026-44410-x. (Introduces the exact GRADCO graphlet-orbit adjacency counter; arXiv:2405.14194.)
> >
> > [13] Ribeiro, P., Paredes, P., Silva, M. E. P., Aparício, D., & Silva, F. (2021). A survey on subgraph counting: concepts, algorithms, and applications to network motifs and graphlets. *ACM Computing Surveys*, 54(2), Article 28, 1–36.

---

> ### Author Response · Authors · 2026-07-03
> **Requested Change 2 and 3**
>
> **Reviewer Comment 2:**
> "Clarify the motivation around linearity. Critical. Distinguish linear semantic relationships in NLP/VLMs from linear separability of node labels in graph embeddings."
>
> **Our response:**
>
> We thank the reviewer for this helpful observation and agree that these are related but distinct notions. In the revised manuscript we will update the introduction to make the connection precise and to distinguish the two explicitly, as follows:
>
> The linear representation hypothesis (LRH) states that meaningful concepts emerge as linear directions or subspaces of the representation space learnt by embedding methods. LHR, which was originally introduced to explain semantic compositionality of word embedding (solving the famous analogy task), was later formalized by Park et al. [1], through three notions (1) subspace representation: a concept corresponds to a linear direction or subspace, (2) measurement: the concept can be recovered by a linear functional probe, and (3) intervention: moving along the direction changes only that concept.
>
> In NLP and VLMs, the LRH is typically evidenced at the semantics level, through compositional vector arithmetic between concepts (the analogy task above). This form of linear relationship is specific to word and multimodal embeddings. For network embeddings, we instead test the LRH through linear separability by using the node class labels as proxies for concepts; a concept is encoded as a linear subspace when the nodes annotated with the corresponding class label are linearly separable from the other nodes, which we formally measure using the performances of a linear classifier.
>
> The evaluation of LHR in NLP/VLM embeddings and in graph embeddings therefore differ in how linearity is tested: compositional arithmetic between concept vectors in NLP/VLMs, versus linear separability of node labels in graph embeddings.
>
> ### References
> [1] Park, Kiho, Yo Joong Choe, and Victor Veitch. "The linear representation hypothesis and the geometry of large language models." arXiv preprint arXiv:2311.03658 (2023).
>
> ---
>
> **Reviewer Comment 3:**
> "Provide theoretical analysis of linear separability. Critical. Add theoretical or formal analysis explaining why more homophilic graphlet-based representations should improve the linear separability of the resulting embedding space."
>
> **Our response:**
>
> We thank the reviewer for raising this point, and we agree that a theoretical analysis would be valuable. Below, we clarify what our contribution establishes, why the question is non-trivial, and what we defer to future theoretical work.
>
> Our study is an empirical one, aimed at understanding when and why network embedding spaces are linearly separable. We show that homophily of the network representation is the key factor: we isolate it from confounders (density, clustering, community structure) both on real networks and in noise-free synthetic networks, and we demonstrate the relationship across 15 networks from multiple domains using a linear SVM. In the revised manuscript we further show that these results are robust to a second linear classifier, shrinkage-LDA, which fits an explicit linear decision boundary in the embedding space.
>
> Regarding linear separability, it is already known that any finite labelled dataset can be made linearly separable given enough dimensions [2], and that with a suitable kernel — for example a Dirac kernel mapping each point to an orthogonal basis vector — every labelling becomes linearly separable by construction [3,4]. The challenge is in fact in designing suitable embedding methods that enable linear separability in the low dimensional spaces typically used for downstream analysis. Our work provides empirical insight into this question and, in doing so, motivates further theoretical research.
>
> To conclude, we characterise this relationship empirically and do not provide a formal proof that higher representation homophily yields higher linear separability. We will state explicitly in the Limitations of our study that our evidence is empirical and not theoritically proven. Establishing the formal link is a natural direction that our empirical results motivate, and we flag it as future work.
>
> ### References
>
> [2] Cover, Thomas M. "Geometrical and statistical properties of systems of linear inequalities with applications in pattern recognition." IEEE transactions on electronic computers 3 (1965): 326-334.
>
> [3] Schölkopf, Bernhard, and Alexander J. Smola. Learning with kernels: support vector machines, regularization, optimization, and beyond. MIT press, 2002.
>
> [4] Shawe-Taylor, John, and Nello Cristianini. Kernel methods for pattern analysis. Cambridge university press, 2004.

---

> ### Author Response · Authors · 2026-07-03
> **Requested Change 4**
>
> **Reviewer Comment 4:**
> "Better position the method against prior work. Critical. Clearly explain how the method differs from higher-order network embeddings, motif-based methods, graphlet Laplacians, GDV-based embeddings, and NetMF-style methods."
>
> **Our response:**
>
> We thank the reviewer for this comment and will add a "Relation to prior work" subsection to the revised manuscript.
>
> Our method produces node embeddings in three steps: (i) we compute the graphlet adjacency matrices $A_{G_k}$, which capture higher-order connectivity by counting how often two nodes co-occur on a given graphlet $G_k$ [5]; (ii) we insert each $A_{G_k}$ into the closed-form random-walk (mutual-information) matrix that DeepWalk, LINE and node2vec were shown to factorise implicitly [6]; and (iii) we factorise the resulting matrix with NMTF. Conceptually, our method bridges higher-order structural representations and random-walk embeddings: rather than summarising higher-order structure per node (as node-level descriptors or pairwise similarity matrices do), we diffuse it across the network through the random-walk formulation.
>
> We now position our method with respect to (1) random-walk, factorization-based approaches (NetMF-style methods), and (2) higher-order structural approaches, including graphlet-spectral embeddings, GDV-similarity based embeddings and the motif-based embeddings of HONE.
>
> **NetMF-style methods (DeepWalk, LINE, node2vec) [5].** Qiu et al. [6] showed that these methods implicitly factorise a closed-form random-walk mutual-information matrix, and made this factorisation explicit via SVD (NetMF). We reuse this closed-form matrix but change both its input and its factorisation: we use the graphlet adjacency instead of the standard adjacency — so the factorised matrix encodes higher-order structure rather than only edge-level proximity — and we factorise it with NMTF instead of SVD.  Unlike SVD, ONMTF returns non-negative factors — giving a parts-based interpretation of the embedding space — and its orthonormality constraint yields independent, non-ambiguous directions that further improve interpretation.
>
> **Graphlet Laplacians / graphlet-spectral embeddings [5].** The graphlet adjacency was introduced as the basis of graphlet-spectral embeddings: a Laplacian is built from a graphlet adjacency, and its eigenvectors give a spectral embedding that generalises Laplacian Eigenmaps. We use the same graphlet-adjacency matrices, but embed them through the random-walk factorisation above rather than a spectral eigendecomposition. In the revised manuscript we add the graphlet-Laplacian / Laplacian-Eigenmaps embedding as a baseline, applied over all nine graphlet adjacencies (not only the standard one), and show that our factorisation-based embeddings outperform the spectral ones.
>
> **GDV / GDV-similarity based embeddings [7].** The graphlet degree vector (GDV) is an 11-dimensional descriptor of the local wiring around a node, counting how many times the node appears in each orbit (its topological position within a graphlet). The GDV-similarity — an orbit-weighted comparison of two GDVs — computed over all node pairs gives the GDV-similarity matrix, a pairwise descriptor of topological similarity. The GDV-PPMI baseline [8] use this GDV-similarity matrix into the DeepWalk closed-form matrix, so the random walks traverse to topologically similar nodes regardelss of whether they are adjacent. The key difference is that GDV-similarity matrix and GDV-PPMI captures topological similarity alone, whereas our graphlet-adjacency representations combine neighbourhood and topological (structural) similarity. Importantly, both the raw GDV-similarity matrix and GDV-PPMI yield less separable embedding spaces than our graphlet-adjacency embeddings.
>
> **Higher-order / motif-based network embeddings — HONE [9].**
> HONE builts on network motifs (subgraphs that are statistically over-represented under a random-graph null model), whereas we use graphlets (induced subgraphs enumerated at any frequency). Beyond this conceptual difference, the key methodological one is how the embedding is generated: HONE builds multiple motif-based matrices and learns a *single* embedding jointly from all of them. The key difference is that HONE pools all motif representations into one embedding, whereas we keep each higher-order graphlet representation $k$ as a separate representation. In our benchmarks the pooled embedding does not outperform our best individual graphlet representation, indicating that pooling compresses the information specific to each graphlet (detailed in our response to Comment 5).

---

> ### Author Response · Authors · 2026-07-03
> **Requested Change 4 References**
>
> ### References
>
> [5] Windels, S. F. L., Malod-Dognin, N., & Pržulj, N. (2019). Graphlet Laplacians for topology–function and topology–disease relationships. *Bioinformatics*, 35(24), 5226–5234.
>
> [6] Qiu, J., Dong, Y., Ma, H., Li, J., Wang, K., & Tang, J. (2018). Network embedding as matrix factorization: Unifying DeepWalk, LINE, PTE, and node2vec. In *Proceedings of the 11th ACM International Conference on Web Search and Data Mining (WSDM)*, 459–467.
>
> [7] Milenković, T., & Pržulj, N. (2008). Uncovering biological network function via graphlet degree signatures. *Cancer Informatics*, 6, 257–273.
>
> [8] Xenos, Alexandros, et al. "Linear functional organization of the omic embedding space." Bioinformatics 37.21 (2021): 3839-3847.
>
> [9] Rossi, R. A., Ahmed, N. K., Koh, E., Kim, S., Rao, A., & Abbasi-Yadkori, Y. (2018). HONE: Higher-Order Network Embeddings. *arXiv preprint* arXiv:1801.09303.

---

> ### Author Response · Authors · 2026-07-03
> **Requested Change 5 Part 1**
>
> **Reviewer Comment 5:**
> "Add stronger baselines. Critical. Include representative higher-order / motif / graphlet embedding baselines, such as HONE or motif-based graph methods."
>
> **Our response:**
>
> We agree with the reviewer that adding another higher-order (i.e., structural) baseline, as HONE (Higher-Order Network Embedding) [9], helps to show the benefits of our method. We further include Laplacian eigenvectors as a baseline, as suggested by Reviewer 6KPH. To make the comparison informative, we apply the Laplacian-eigenvector baseline not only to the standard adjacency but over all nine graphlet adjacencies $A_{G_k}$ ($k=0..8$, where $k{=}0$ is the standard adjacency), to test whether higher-order structure improves this baseline as well (closely related to the Graphlet Laplacians [6]). Finally, as requested by Reviewer bu8G, we add three GNNs (GCN, GAT and Motif-GNN) that are trained to predict node class labels from both the graph structure and additional node features. As a result, the GNNs have an advantage over the purely structural approaches whenever the additional node features are predictive of the class labels.
>
> We compare the GNNs and the structural baselines on the node-classification task; the structural baselines are evaluated by training an L-SVM / RF classifier on their embeddings, whereas the GNNs are trained end-to-end. Across these extended benchmarks we observe that on the small filtered heterophilic networks (Chameleon, Squirrel) no method dominates: the two families are comparable, with the purely structural Laplacian winning on Chameleon and Motif-GNN on Squirrel. GNNs win where the signal is in the node features (Roman-empire), and our graphlet embeddings win where it is in the network structure (Amazon-ratings). On the feature-free biological networks our graphlet embeddings with a linear classifier beat all three GNNs and the HONE baseline on every network (full tables in our response to Comment 2 of Reviewer 6KPH).
>
> We also compare the embedding spaces produced by the structural baselines on the downstream analysis tasks: (i) information-retrieval on the single-label networks and (ii) on the functional module discovery on the biological networks.
>
> *Cosine-weighted AUROC for the embedding methods (downstream retrieval task); best over $k$ with the corresponding $k$ in parentheses (HONE reports the best variant); best per network in bold.*
>
> | Network | DeepWalk$_{G_k}$ | LINE$_{G_k}$ | $A_{G_k}$ | Laplacian | HONE |
> |---|---|---|---|---|---|
> | Cora | **0.782 (k=3)** | 0.722 (k=3) | 0.726 (k=3) | 0.744 (k=1) | 0.740 |
> | CiteSeer | 0.650 (k=3) | 0.611 (k=3) | **0.654 (k=3)** | 0.653 (k=0) | 0.618 |
> | Wikipedia CS | **0.801 (k=0)** | 0.765 (k=3) | 0.715 (k=1) | 0.729 (k=0) | 0.773 |
> | CS Co-author | **0.880 (k=0)** | 0.841 (k=3) | 0.827 (k=1) | 0.831 (k=0) | 0.846 |
> | USA air-traffic | 0.594 (k=8) | 0.567 (k=8) | **0.596 (k=7)** | 0.545 (k=0) | 0.549 |
> | Chameleon-filt. | **0.544 (k=4)** | 0.533 (k=0) | 0.540 (k=8) | 0.541 (k=6) | 0.538 |
> | Squirrel-filt. | **0.521 (k=3)** | 0.513 (k=0) | 0.510 (k=4) | 0.519 (k=3) | 0.509 |
> | Amazon-ratings | 0.512 (k=0) | 0.512 (k=4) | 0.516 (k=5) | **0.521 (k=0)** | 0.512 |
> | **Average** | **0.660** | 0.633 | 0.636 | 0.635 | 0.636 |
>
> On the information-retrieval task, our graphlet-based extension of the DeepWalk embeddings (DeepWalk${G_k}$) achieves the highest average AUROC (0.660) and is the best method on 5 of 8 networks, ahead of both the Laplacian baseline (0.635) and HONE (0.636); on two of the three remaining networks, the best result is obtained by another graphlet-based embedding ($A{G_k}$, on CiteSeer and USA air-traffic), and only on Amazon-ratings — where all methods are on par (AUROC $\approx$ 0.51–0.52) — does the Laplacian baseline edge marginally ahead. We further observe that the Laplacian baseline itself improves under our graphlet-based extensions: its best result is obtained at a higher order ($k>0$) on several networks (e.g. Cora $k=1$, Chameleon $k=6$, Squirrel $k=3$), so higher-order structural information benefits even a standard spectral method. Finally, HONE — which learns a single embedding jointly from all motif adjacencies — does not outperform our individual graphlet-based representations, indicating that pooling all higher-order information into one embedding does not, on its own, surpass the selection of an appropriate graphlet representation, and thereby reinforcing the motivation for our approach.

---

> > ### Author Response · Authors · 2026-07-03
> > **Requested Change 5 Part 2**
> >
> > Finally, we also compare the structural baselines with our methods on the **functional module discovery** task on the six biological networks. Recall that in this downstream task we first cluster the genes in the embedding space (KMeans, $\sqrt{N/2}$ clusters) and test each cluster (module) for enrichment of Reactome pathways; *functional coverage* is the fraction of annotated Reactome pathways that are significantly enriched in at least one module, i.e. how much of the known biology the embedding organises into coherent modules.
> >
> > *Reactome functional coverage (%) from the discovered modules; best over $k$ with the corresponding $k$ in parentheses (HONE reports the best variant); best per network in bold.*
> >
> > | Network | DeepWalk$_{G_k}$ | LINE$_{G_k}$ | $A_{G_k}$ | Laplacian | HONE |
> > |---|---|---|---|---|---|
> > | Pombe COEX | **48.5 (k=0)** | 43.6 (k=1) | 41.5 (k=4) | 45.6 (k=7) | 39.5 |
> > | Cerevisiae COEX | **63.3 (k=7)** | 62.1 (k=1) | 58.3 (k=5) | 59.1 (k=1) | 47.1 |
> > | Homo sapiens COEX | 61.5 (k=2) | 65.6 (k=5) | 63.9 (k=8) | **66.1 (k=7)** | 54.5 |
> > | Pombe PPI | **50.6 (k=0)** | 33.1 (k=3) | 25.6 (k=8) | 36.5 (k=5) | 34.5 |
> > | Cerevisiae PPI | **57.3 (k=2)** | 45.6 (k=0) | 34.0 (k=2) | 52.2 (k=0) | 49.6 |
> > | Homo sapiens PPI | **73.6 (k=8)** | 69.7 (k=8) | 63.8 (k=8) | 52.3 (k=2) | 62.7 |
> > | **Average** | **59.1** | 53.3 | 47.9 | 52.0 | 48.0 |
> >
> > On this downstream task, our graphlet-based extensions of DeepWalk (DeepWalk$_{G_k}$) captures the most biological signal: it has the highest functional coverage on five of the six networks and the highest average (59.1\%), clearly outperforming both the Laplacian (52.0\%) and HONE (48.0\%) baselines.

---

> ### Author Response · Authors · 2026-07-03
> **Requested Change 6**
>
> **Reviewer Comment 6:**
> "Address graphlet selection bias. Critical. Provide a principled rule for selecting graphlet representations, or use a strict validation/nested cross-validation protocol. Report fixed, average, and best graphlet results separately."
>
> **Our response:**
>
> We thank the reviewer for raising the graphlet-selection question and we address it on fourth levels. Please note that the same point was raised by Reviewer bu8G; for ease of reading, we reproduce our response here.
>
> First, we would like to clarify that homophily is the explanatory variable in our study — it accounts for why an embedding space is linearly separable — not a quantity we ask practitioners to compute to use the method. Our scientific claim (homophily is associated with separability) is distinct from the practical question of choosing a graphlet-based representation.
>
> Second, we agree with the reviewer that the selection of the corresponding graphlet representation is challenging. Please also note that embedding methods are all heuristics that are bound to fail on some instances (i.e., there is no “best for all”). For that reason, in practice, we rely on domain expertise to select the best graphlet representation. For instance, in social networks it is known that the triadic closure is the most frequent graph substructure [10] and in molecular networks that interecting genes form interconnected modules corresponding to dense graphlets (G2 and G8) [11].
>
> Third, we introduce representation coverage (fraction of nodes in the largest connected component of a representation), a purely structural, label-free filter that discards fragmented high-order graphlet representations (those retaining fewer than half of the nodes) before any labels are seen.
>
> Fourth, pooling all graphlets into one embedding does not bypass the selection problem. Recall that we have already shown that factorizing the GDV similarity matrix or the GDV-PPMI matrix yields less separable embedding spaces than our new graphlet-based methods. To further test this, we add as an additional baseline HONE (Higher-Order Network Embedding), which learns jointly from all motif adjacencies a single embedding, so no graphlet need to be chosen. Across our downstream tasks HONE is matched or outperformed by an individual graphlet-based representation, confirming that concatenating all motifs into one embedding does not resolve the selection problem.
>
> In the revised version of the manuscript, we will add as a limitation that we do not provide an automatic, fully label-free rule for selecting the single best graphlet representation, instead we rely on domain knowledge together with the coverage filter.
>
> On reporting and validation: we now report the **fixed** (G0, the standard adjacency), **average** (mean over k), and **best** (max over k) graphlet results separately in all tables and figures, so the gain from higher-order graphlet representation is transparent and the no-selection baseline is visible.
>
> ### References
>
> [10] Bianconi, Ginestra, et al. "Triadic closure as a basic generating mechanism of communities in complex networks." arXiv preprint arXiv:1407.1664 (2014).
>
> [11] Sharan, Roded, Igor Ulitsky, and Ron Shamir. "Network‐based prediction of protein function." Molecular systems biology 3.1 (2007)

---

> ### Author Response · Authors · 2026-07-03
> **Requested Change 8**
>
> **Reviewer Comment 8:**
> "Clarify assumptions and limitations. Critical. Discuss when local graphlet/motif structures are expected to correlate with labels, and when the method may fail."
>
> **Our response:**
>
> We thank the reviewer for the comment. We now state the method's core assumption explicitly, ground it in prior evidence, and characterise the cases in which it is expected to hold and to fail, supported by our extended benchmarks.
>
> **Our assumption:** higher-order, graphlet based matrix representation of networks will be more homophilic than baseline $k=0$ matrix representation when there exist topology-function relationships in the networks, i.e., associations between the local topology of the nodes (as captured by higher-order graph substructures such as graphlets) and the nodes’ functions (class labels, biological annotations of the nodes). As our correlation analysis suggests, these more homophilic representations then lead to more linearly separable embedding spaces, which are more suited for downstream analysis tasks.
>
> **When the assumption holds, and when it fails.** We expect our graphlet-based extension to perform better thasn baseline ($k=0$) when such topology–function relationships are present. Notably, such relationships have been observed accross domains: in biological networks [14], social networks (e.g., triadic closure) and economic networks (e.g., world trade networks) [15]. Note that for higher order topology to exist, the networks need minimum edge density [16], so we expect our methodology to fail on very sparse networks. In addition, even when structure exists, our method fails when node labels are not correlated with local wiring: either because labels are unrelated to topology (e.g. randomly wired networks) or because all nodes share near-identical local structure and are therefore indistinguishable (e.g. regular, grid-like networks).
>
> **Empirical demonstration of when the method works/fails.** On the small filtered datasets (Chameleon, Squirrel) no method dominates: the best result is split between the structure-only methods and the GNNs, which are trained not only on the graph structure but also on additional node features. The two additional heterophilic benchmarks are very informative: Roman-empire is our failure case — extremely sparse (density = 0.00013) with essentially no homophily (node homophily ≈ 0.046), so our structure-only embeddings are near-random (F1 score 0.13–0.18) while feature-using models recover the labels from features (0.53–0.61); on Amazon-ratings, where the local topology of the nodes reflect their labels, our structure only methods (F1 score of 0.52) instead outperform all three GNNs that uses both structure and additional node features (F1 scores in between 0.23 and 0.40). On the feature-free biological networks, where no node feature is available and both our method and the GNNs use only the network structure, there is a graphlet-based embedding with a linear classifier that outperforms all three GNNs on every network. In short, our method succeeds when network structure is predictive of the node labels and fails where it does not.
>
> To sum up: when the network is too sparse for higher-order graphlets to exist and/or the additional node features are more predictive of the node labels than the network structure, GNNs outperform the feature-free methods. When features are absent or uninformative and the label information is instead associated with the network structure — as in molecular networks, transport networks, and many other real-world graphs — purely structural graphlet-based embeddings with a simple linear classifier match or outperform end-to-end GNNs. The clear failure mode is very sparse networks (e.g. Roman-empire), where higher-order graph structures are not formed. We will state this assumption and its failure mode explicitly in the revised manuscript.
>
> ### References
>
> [14] Davis, D., Yaveroğlu, Ö. N., Malod-Dognin, N., Stojmirovic, A., & Pržulj, N. (2015). Topology-function conservation in protein–protein interaction networks. *Bioinformatics*, 31(10), 1632–1639.
>
> [15] Yaveroğlu, Ömer Nebil, et al. "Revealing the hidden language of complex networks." Scientific reports 4.1 (2014): 4547.
>
> [16] Wayne Hayes, Kai Sun, Nataša Pržulj, Graphlet-based measures are suitable for biological network comparison, Bioinformatics, Volume 29, Issue 4, February 2013, Pages 483–491

---

> ### Author Response · Authors · 2026-07-03
> **Requested Change 9**
>
> **Reviewer Comment 9:**
> "Deepen the empirical analysis of linear separability."
>
> **Our response:**
> We thank the reviewer for the suggestion and add a direct analysis of the embedding geometry. For every embedding space we compute the **inter-/intra-class distance ratio**: the mean Euclidean distance between nodes of different classes divided by the mean distance between nodes of the same class. A ratio $>1$ means that, on average, two nodes from different classes are farther apart than two nodes from the same class; a larger ratio indicates higher between-class separation. We report the best value over graphlet representation $k$ ($k$ in parentheses), against the Laplacian and HONE embedding baselines. Please note that we only perform inter/intra class distance based analysis on single label networks as it is ill-defined/not suited for the overlapping classes of our multi-label networks.
>
> *Table — single-label networks: inter-/intra-class Euclidean distance ratio (best over $k$, corresponding $k$ in parentheses; HONE reports the best variant); best per network in bold.*
>
> | Network | DeepWalk$_{G_k}$ | LINE$_{G_k}$ | $A_{G_k}$ | Laplacian | HONE |
> |---|---|---|---|---|---|
> | Cora | **1.267 (k=3)** | 1.106 (k=3) | 1.233 (k=3) | 1.077 (k=6) | 1.007 |
> | CiteSeer | **1.370 (k=3)** | 1.084 (k=3) | 1.050 (k=3) | 1.248 (k=3) | 1.041 |
> | Wikipedia CS | **1.221 (k=6)** | 1.175 (k=3) | 1.094 (k=3) | 1.080 (k=1) | 1.179 |
> | CS Co-author | **1.434 (k=3)** | 1.107 (k=3) | 1.222 (k=3) | 1.202 (k=2) | 1.127 |
> | USA air-traffic | 1.191 (k=5) | 1.179 (k=5) | **1.520 (k=5)** | 1.112 (k=5) | 1.125 |
> | Chameleon-filt. | 1.130 (k=4) | 1.090 (k=3) | 1.101 (k=2) | **1.131 (k=0)** | 1.085 |
> | Squirrel-filt. | 1.186 (k=5) | 1.107 (k=8) | 1.110 (k=8) | **1.197 (k=5)** | 1.086 |
> | Roman-empire | 1.178 (k=2) | 1.116 (k=2) | 1.153 (k=2) | 1.058 (k=2) | **1.210** |
> | Amazon-ratings | 1.008 (k=3) | 0.992 (k=3) | 0.983 (k=2) | **1.011 (k=4)** | 0.981 |
>
> These results suggest that
>
> **(1) Class separation appears only where structural signal is present.** On the four homophilic networks (Cora, CiteSeer, Wikipedia-CS, CS Co-author), our graphlet-extensions of DeepWalk (DeepWalk$_{G_k}$) has the largest intra-class distance ratio in every network (up to 1.43). On the heterophilic networks, by contrast, no representation separates the classes: ratios stay near 1 (Amazon-ratings $\approx 1.0$; Chameleon/Squirrel/Roman-empire $\approx 1.1$–1.2), with the differences between methods being marginal. Roman-empire is the sparsest network in our benchmark (density = 0.00013); at this density higher-order graphlets rarely appear, so the individual graphlet representations yield low ratios (1.06–1.18). The only exception is HONE, which factorises all motif representations jointly rather than one at a time; by combining all representations, it leads to a marginally higher inter-/intra-class distance ratio (1.21). The geometry thus follows the aforementioned topology–function relationship: separation emerges only where the network topology is associated with the node labels.
>
> **(2) Where separation exists, it is driven by higher-order structure.** The highest intra-class distance is almost never yielded by the standard adjacency ($G_0$), but by a higher-order graphlet representation. To quantify this across all methods, the table below reports, for each embedding family, how often its highest inter-/intra-class distance ratio is reached at $G_0$ versus at a higher-order graphlet representation, together with the average improvement over the inter-/intra-class distance ratio observed on the standard adjacency ($G_0$).
>
> *Per-family count on the nine single-label networks: number on which each family's largest inter-/intra-class ratio is reached at $G_0$ ($k=0$) vs at a higher order ($k>0$); last column gives the average increase of that best ratio over the family's own $G_0$.*
>
> | Embedding family | best at $G_0$ ($k=0$) | best at higher order ($k>0$) | avg. ratio gain over $G_0$ |
> |---|---|---|---|
> | DeepWalk$_{G_k}$ | 0 | **9** | +0.11 |
> | LINE$_{G_k}$ | 0 | **9** | +0.07 |
> | $A_{G_k}$ (graphlet adjacency) | 0 | **9** | +0.09 |
> | Laplacian (over $A_{G_k}$) | 1 | **8** | +0.05 |
>
> The highest inter-/intra-class distance ratio is reached at $k>0$ for every family and essentially every network (9/9 for all three graphlet embeddings, 8/9 for the Laplacian baseline), with the ratio improving by up to $+0.11$ (DeepWalk$_{G_k}$) over the standard adjacency. This is the geometric counterpart of our classification finding: higher-order graphlet structural embeddings improve the F1-score by widening the inter-class distances relative to the intra-class ones.
>
> Finally, following Reviewer's 6KPH comment, we demonstrate robustness to different splits by reporting the mean $\pm$ std throughout our $F_1$ tables: every result is averaged over the official ten train/test splits for the single-label networks, and ten random seeds for the feature-free biological networks.

---

> ### Comment · Reviewer_g9af · 2026-07-17
>
> Thank you for the detailed response and the substantial additional work. I appreciate the authors’ efforts to narrow the scope, add stronger baselines, correct the heterophilic benchmarks, report repeated-split results, and discuss the assumptions and limitations. These revisions improve the empirical completeness of the paper.
>
> However, two central concerns remain. First, the main claim regarding linearly separable embedding spaces is still insufficiently established. The additional classifiers, correlation analyses, and inter-/intra-class distance statistics provide useful empirical evidence, but they do not explain why increased representation homophily should lead to improved linear separability. Since this relationship is the core contribution of the paper, I believe a theoretical or formal analysis is needed to clarify the conditions, assumptions, and limitations under which it holds. Linear-probe performance and an empirical F1 threshold alone are not equivalent to geometric linear separability.
>
> Second, the graphlet-selection problem remains unresolved. The coverage filter only removes fragmented representations and does not provide a principled rule for selecting among the remaining graphlets. Reliance on domain knowledge is not a general or reproducible selection strategy, while reporting the best result over graphlets remains an oracle-style evaluation. In addition, I do not find the newly added fixed, average, and best results, it may separately improves transparency but does not eliminate the selection bias or provide a practical label-free solution. I would suggest to find out the connection between different setting or try to find a robust ensemble.
>
> Overall, although the revision strengthens the submission, these two issues continue to limit both the validity of the central claim and the practical value of the proposed framework. I therefore maintain my previous recommendation.

---

> ### Comment · Reviewer_6KPH · 2026-07-17
>
> I just wanted to weigh in and note that the paper does not claim a theoretical contribution. As stated in the TMLR guidelines, our primary task is to assess whether the claims actually made by the authors are supported by accurate, convincing, and clear evidence.
>
> I agree that the wording around “linear separability” should be carefully scoped, since the paper provides empirical rather than formal support. However, if the authors frame the result as an empirical relationship and provide sufficiently clear and convincing evidence across appropriate datasets and settings, I do not think the absence of a theorem should itself be treated as a reason for rejection. Requiring a theoretical result here would amount to requesting a different type of contribution from the one the paper presents. This seems especially relevant under TMLR’s evaluation criteria, which focus on the validity and interest of the submitted claims rather than requiring every paper to satisfy a conference-style template of theoretical novelty.

---

> > ### Comment · Reviewer_g9af · 2026-07-18
> >
> > Thank you for reminding the  evaluation criteria. Regardless the theoretical part, I would like to see the performance of fixed (G0, the standard adjacency), average (mean over k), and best (max over k) graphlet results. Unfortunately, I cannot find the newest version provided by the authors.  If the performance of "fixed" is relatively stable, I would say this paper provide a solid empirical study on the graphlet representation.
> >
> > And once again, the authors should carefully discuss the “linear separability” and their evaluation methods.